# Learning to Remove, Not Repeat: Robust Object Removal in Cluttered Scenes using Diffusion Models

## Abstract

Object removal, a key image inpainting task, aims to erase specified objects and plausibly fill the resulting region. Although recent diffusion models excel at generating realistic content, when employed for the removal task, they often fail in cluttered scenes by replicating nearby objects or hallucinating semantically similar ones, an artifact of their powerful, yet context-agnostic, generative priors. To address this, we introduce a robust framework that Learns to Remove, Not Repeat (LRNR). Our approach has three key components. First, we propose the Scatter-Tile Object Removal (STORe) dataset, a large-scale synthetic dataset with unique scatter and tile configurations designed to make models robust to object replication. Second, we employ an efficient fine-tuning strategy that combines Low-Rank Adaptation (LoRA) with a learnable task prompt, which internalizes the concept of removal, thereby eliminating the need for manual text guidance. Third, we introduce Mask-Aware Scheduled Guidance (MASG), a training-free inference technique that spatially and temporally modulates classifier-free guidance to enhance inpainting quality and preserve background integrity. Our evaluations demonstrate that LRNR outperforms state-of-the-art approaches, particularly in terms of removal success rate in challenging scenes prone to object replication, leading to more reliable and semantically correct results. Our dataset, source code, and trained models will be publicly available.

## 1 Introduction

The ability to edit digital images with high fidelity is increasingly essential across a broad range of applications—from professional photo and video editing to everyday social media use Bar-Tal et al. (2022). A fundamental task in this realm is image inpainting, which aims to fill missing or corrupted regions within an image with plausible and visually coherent content Jampani et al. (2021); Liu et al. (2025). Object Removal is a subtask of image inpainting that aims to remove the masked object within the input image. It requires both (i) accurate elimination of the target object without introducing artifacts or novel, unwanted elements, and (ii) seamless restoration of the masked region, maintaining consistency in texture, lighting, and overall scene structure Liu et al. (2025).

Historically, object removal has evolved from patch-based methods Barnes et al. (2009), which copy content from surrounding regions, to learning-based approaches that synthesize content based on semantic understanding. In addition, CNN and Transformer-based methods Yan et al. (2018); Oleksii (2019); Dong et al. (2022); Cao et al. (2023) improved contextual awareness, while GAN-based techniques Goodfellow et al. (2014); Suvorov et al. (2021); Zhao et al. (2021); Li et al. (2022b) improved realism through adversarial training. Despite their progress, these methods often suffer from ghosting artifacts, blurry reconstruction, or replication of nearby structures. CNN-based methods are limited by their fixed receptive fields Suvorov et al. (2021), and GANs remain difficult to train and prone to mode collapse Salimans et al. (2016).

Diffusion models Ho et al. (2020) have recently become the state-of-the-art in generative modeling, particularly in image synthesis, by iteratively denoising random noise into coherent samples. Inpainting models based on diffusion Zhuang et al. (2023); Winter et al. (2024); Li et al. (2025); Sun

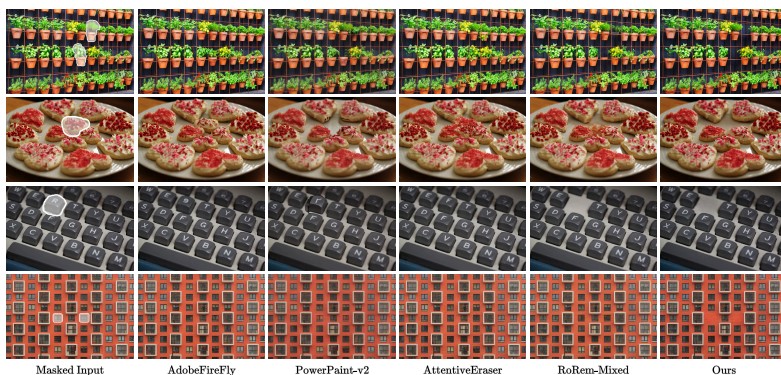

Figure 1: **LRNR** is the first object removal method that is robust in not replicating similar objects from adjacent scenes. Other state-of-the-art methods, AdobeFireFly ado (2025), PowerPaint Zhuang et al. (2023), AttentiveEraser Sun et al. (2025), and RoRem Li et al. (2025) tend to generate similar objects or struggle to remove the masked object completely, leading to unrealistic output.

et al. (2025); Liu et al. (2025); Ju et al. (2024) leverage strong generative priors and conditioning inputs (e.g., masks, prompts) to produce high-quality, semantically plausible results. These capabilities make them especially effective for complex image manipulation tasks such as object removal. However, naively applying diffusion models to object removal presents several challenges. Standard pipelines are often trained on datasets with random masks and optimized for general inpainting rather than explicit object elimination, which, despite their impressive generative capabilities, often results in poor object removal performance and a strong dependence on carefully crafted prompts to guide the model toward successful erasure.

To address these limitations, some prior works Li et al. (2025); Winter et al. (2024); Li et al. (2024); Sagong et al. (2022); Liu et al. (2025) have adopted triplet-based training—using *source*, *mask*, and *target* image tuples—to better align the generative process with the object removal objective. While such approaches improve remove performance, they still often regenerate semantically similar objects or introduce random, incongruent content into the masked region, particularly in cluttered scenes with multiple similar instances, unless explicitly constrained toward erasure (see Figure 1).

To address this, we introduce **STORe (Scatter-Tile Object Removal)**, a new synthetic dataset designed to challenge models with densely packed, visually similar objects. STORe is constructed from and extends existing datasets such as RORD Sagong et al. (2022), Mulan Tudosiu et al. (2024), and HQ-SAM Ke et al. (2023), and encourages models to perform precise object removal by disentangling similar-looking structures across the image.

Inspired by PowerPaint Zhuang et al. (2023), we define a learnable task prompt $P_{\text{rmv}}$ that encodes the removal objective. During training, this prompt guides the model to reconstruct background content instead of replicating the masked object or similar elements from the surrounding scene. At inference time, object removal is achieved simply by providing $P_{\text{rmv}}$, without any user-defined prompt.

To further improve image quality and minimize background degradation, we propose **Mask-Aware Scheduled Guidance (MASG)**, a training-free inference strategy that applies different levels of Classifier-Free Guidance (CFG) to masked and unmasked regions. By assigning a high guidance scale to the masked area, MASG intensifies the model's focus on the task prompt $P_{\text{rmv}}$, reinforcing the objective of object removal. Simultaneously, a lower guidance scale is applied to unmasked regions to preserve contextual fidelity. This region-specific guidance enables MASG to generate coherent and more realistic outputs by enhancing task-specific synthesis within the mask while maintaining the integrity of the surrounding background.

## 2 RELATED WORKS

### 2.1 FOUNDATIONS OF IMAGE INPAINTING

**GAN-based Methods.** Early inpainting models combined CNNs and GANs, such as Context Encoders Pathak et al. (2016), which conditioned hole-filling on surrounding context. LaMa Suvorov

et al. (2021) extended this with Fast Fourier convolutions Nair et al. (2020) to support larger missing areas. While these models perform reasonably well on repetitive textures (e.g., sky, grass), they often struggle on complex scenes due to poor global context modeling and blurry fills.

**Diffusion-based Methods.** Diffusion-based models Ho et al. (2020) represent a significant advancement in generative modeling. Pixel-based methods like RePaint Lugmayr et al. (2022) employ iterative denoising for masked regions, while latent-space models like Stable Diffusion Inpainting Rombach et al. (2021b) produce high-resolution, coherent reconstructions. Further developments, such as Blended Diffusion Avrahami et al. (2022) and Dream-Inpainter Xie et al. (2023b), incorporate text and subject conditioning. Despite improvements, these methods may misalign fine details or hallucinate undesired content.

## 2.2 OBJECT REMOVAL VIA DIFFUSION

**Training-based Methods.** Many works fine-tune diffusion models on removal or editing tasks, like PowerPaint Zhuang et al. (2023), introduce some special tokens so the model can switch between tasks. Further, InstructDiffusion Geng et al. (2024) frames diverse editing tasks as instruction-following by fine-tuning on large instruction datasets, and RoRem Li et al. (2025) fine-tunes Stable Diffusion XL Podell et al. (2023) on a semi-automatic "human-in-the-loop" dataset. These improve removal accuracy but require costly data or training pipelines.

**Inversion-based Methods.** Some methods first invert the input image into the diffusion latent and then apply edits. Null-Text Inversion Mokady et al. (2023) reconstructs the input image and allows prompt-guided edits without retraining. These methods preserve input fidelity but require careful optimization.

**Attention/Semantic-Guided Control.** Beyond the above, some works explicitly integrate semantic cues or attention masks to improve quality. Attentive Eraser Sun et al. (2025), CLIPAway Ekin et al. (2024), and MagicEraser Li et al. (2024) explicitly steer attention or use CLIP-based embeddings to avoid regenerating removed content, achieving semantically aware inpainting.

## 2.3 DATASETS AND DATA-GENERATION STRATEGIES

Training data for inpainting models is developed through three main strategies. Self-supervised masking on large datasets like COCO Lin et al. (2014) is a common approach, but it often fails to capture realistic object shape priors. To enable text-driven removal, synthetic prompt-oriented pairs are generated using LLMs and diffusion models, creating scalable instruction-aligned datasets like InstructPix2Pix Brooks et al. (2022) and Inst-Inpaint Yildirim et al. (2023), though these can suffer from synthetic artifacts. The most reliable supervision comes from real-world paired data, such as the RORD dataset Sagong et al. (2022), which provides precise ground truth by capturing scenes before and after object removal. However, this data is costly to produce. Consequently, a practical strategy often involves combining scalable synthetic data with high-quality, realistic pairs to achieve robust background reconstruction.

# 3 METHODOLOGY

## 3.1 PROBLEM FORMULATION

The object removal task is formulated as a conditional image generation problem. Let $\mathbf{x}_s$ denote the source image, which contains an unwanted object, and let $\mathbf{M}$ be a corresponding binary mask indicating the object's location, where '1's denote the pixels belonging to the object to be removed, and '0's denote the background. The objective is to generate an edited image $\mathbf{x}_e$, in which the region corresponding to the mask $\mathbf{M}$ has been filled with visually and semantically coherent content that blends naturally with the surrounding context. The model $G_\theta$ must therefore learn a mapping that takes the source image and mask as input and produces a plausible completion, effectively learning the distribution $p(\mathbf{x}_e|\mathbf{x}_s, \mathbf{M})$. The unmasked portion of the image, which provides the visual context, can be represented as $\mathbf{x}_s \odot (1 - \mathbf{M})$, where $\odot$ denotes element-wise multiplication, and should be untouched during the generation process.

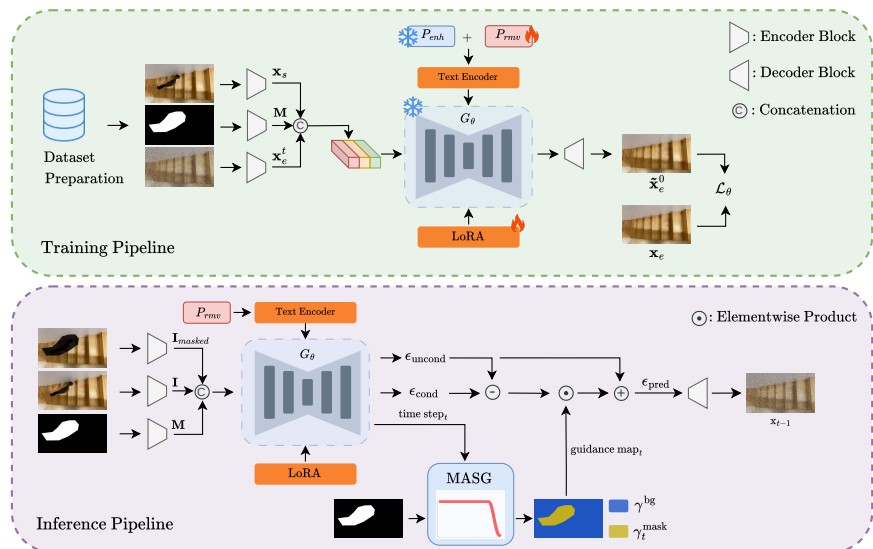

Figure 2: Overview of our fine-tuning pipeline. We concatenate the source image, mask and the noisy edited image as the model input. During training, only the text encoder's parameters and the LoRA weights are trained and rest of the parameters will be frozen.

## 3.2 STORe DATASET

A primary challenge in object removal is erasing a target from a group of similar instances. Even state-of-the-art models like RoRem Li et al. (2025) often fail in these cluttered scenarios, as shown in Figure 1, by regenerating the object they are meant to remove. This widespread failure stems from a fundamental gap in training data. Current datasets are inadequate, as they are either synthetically generated using these same flawed models or consist of costly, simplistic real-world captures like RORD Sagong et al. (2022). Critically, no existing dataset is designed to train models to remove a single object from a group of near-duplicates. To bridge this gap, we introduce **STORe**, a novel, large-scale synthetic dataset explicitly created to address this challenge without the high cost and scalability limitations of manual annotation. STORe is built from 2,700 curated scenarios from RORD and 44,000 object instances from HQ-SAM Ke et al. (2023), using two complementary construction strategies: the Tile Method and the Scatter Method. Figure 3 presents sample images from the STORe dataset.

The **Tile Method** generates extreme cases of object repetition by tiling a source image into a grid of identical instances. A random tile is then replaced with its background to create the training pair. This approach excels at creating structured, high-density scenes for stress-testing models. However, it has key limitations: the rigid grid prevents augmentations like rotation or scaling, repetitive background patterns can introduce bias, and minor edge artifacts may remain despite blending, reducing scene naturalism.

To overcome the Tile Method's limitations, the **Scatter Method** enhances scene realism and diversity. We programmatically "scatter" multiple object instances from HQ-SAM onto clean RORD backgrounds, applying random augmentations in scale, rotation, and position to simulate real-world clutter. One object is then removed to create the target pair, with a blending pipeline ensuring seamless integration. This method generates varied, unstructured compositions that compel the model to learn nuanced object-context relationships rather than fixed patterns. By combining the Scatter Method's realism with the Tile Method's stress-testing, STORe provides a comprehensive training curriculum.

## 3.3 TRAINING STRATEGY

We adopt a triplet-based fine-tuning strategy using $(\mathbf{x}_s, \mathbf{M}, \mathbf{x}_e)$, where $\mathbf{x}_s$ is the source image containing the object, $\mathbf{M}$ is the binary mask indicating the object region, and $\mathbf{x}_e$ is the edited image with the object removed. Following Li et al. (2025), we use the edited image $\mathbf{x}_e$ as the generation

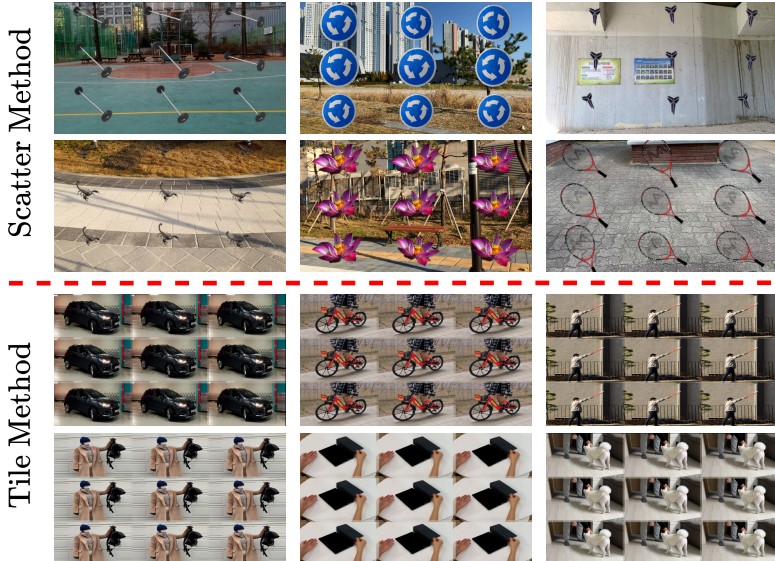

Figure 3: some samples from the **STORe** dataset. (1) Tile method uses large-mask RORD frames, tiles them, replaces one tile with its ground-truth removal, and blends seams; (2) Scatter method pastes the HQ-SAM object patches onto RORD backgrounds with random augmentations, then removes one object per image to form data triplets.

target and condition the model on both the source image and mask. Although RoRem suggests that conditioning on $\mathbf{x}_s$ may result in residual artifacts, we observe improved learning by exposing the model to both the pre- and post-removal states, as supported by ObjectDrop Winter et al. (2024).

We fine-tune the SDXL Inpainting model for object removal using a parameter-efficient setup. The base model weights $\theta$ are frozen, and LoRA is applied. This design significantly reduces memory consumption while mitigating the risk of catastrophic forgetting Hu et al. (2022).

We further guide the model using a learnable object removal prompt, denoted as $P_{\text{rmv}}$, inspired by PowerPaint Zhuang et al. (2023), whose embedding $v^*_{\text{rmv}}$ is optimized during training. This token is embedded into the CLIP text encoder $\tau_\gamma$ and is designed to capture the semantics of object removal across diverse visual scenarios. To maintain output fidelity, we augment this dynamic task token with a fixed set of quality-preserving concepts known to guide diffusion models toward visually consistent outputs. These fixed tokens remain frozen during training. The final composite prompt is defined as $P_{\text{final}} = P_{\text{enh}} + P_{\text{rmv}}$, where $P_{\text{enh}}$ represents the set of static enhancement tokens. This formulation enables $v^*_{\text{rmv}}$ to specialize in learning object removal behavior without being burdened by quality preservation, thus improving both task accuracy and visual coherence in generated outputs (see Appendix I for list of $P_{enh}$ words).

During training, the model is tasked with denoising a perturbed version of the edited image $\mathbf{x}^t_e = \alpha_t \cdot \mathbf{x}_e + \sigma_t \cdot \epsilon$, where $t \sim [0, T]$, $\epsilon \sim \mathcal{N}(0, \mathbf{I})$, and $\alpha_t, \sigma_t$ are determined by the noise scheduler. The input to the model is the concatenation $[\mathbf{x}^t_e, \mathbf{M}, \mathbf{x}_s]$, and the objective is to predict the added noise:

$$\min_{\phi, v^*} \mathbb{E}_{\mathbf{x}_e, \mathbf{M}, t, \epsilon_t} \left[ \left\| \epsilon_t - G_{\theta, \phi} \left( [\mathbf{x}^t_e, \mathbf{M}, \mathbf{x}_s], \tau_{\omega'}(P_{\text{final}}), t \right) \right\|^2_2 \right], \tag{1}$$

where $\phi$ represents the LoRA parameters and $\omega' = \{v^*_{\text{rmv}}\} \cup \omega$ denotes the augmented text encoder parameters, with $\omega$ frozen and only $v^*_{\text{rmv}}$ optimized during training.

### 3.4 MASK-AWARE SCHEDULED GUIDANCE

Fine-tuning large-scale inpainting models like SDXL on limited datasets poses a significant risk of overfitting. While employing a high guidance scale for the removal prompt ($P_{\text{rmv}}$) within the CFG framework Ho & Salimans (2022) enhances object removal, it often introduces undesirable artifacts such as color bleeding and structural inconsistencies in the inpainted regions. To mitigate these issues, we propose MASG, a novel training-free approach that modulates guidance scale spatially

---

**Algorithm 1** Denoising with MASG

---

**Input:** Binary mask $\mathbf{M}$, prompt $P_{\text{final}}$
**Params:** SDXL model $G_{\theta,\phi}$, timesteps $T$, sampler $S$, $\gamma^{\text{bg}}$, $\gamma_T^{\text{mask}}$, cutoff $t_c$
**Output:** Final latent $\hat{\mathbf{x}}_0$

1: $\mathbf{M}' \leftarrow \text{Downscale}(\mathbf{M})$, $\mathbf{x}_T \sim \mathcal{N}(0, I)$
2: **for** $t = T, \ldots, 1$ **do**
3:     Compute $\gamma_t^{\text{mask}}$ via Eq. equation 3
4:     $\mathbf{M}_{\text{cfg}} \leftarrow \gamma^{\text{bg}}(1 - \mathbf{M}') + \gamma_t^{\text{mask}} \cdot \mathbf{M}'$
5:     $\epsilon_{\text{uncond}} \leftarrow G_{\theta,\phi}([\mathbf{x}_e^t, \mathbf{M}', \mathbf{x}_s], t, \varnothing)$
6:     $\epsilon_{\text{cond}} \leftarrow G_{\theta,\phi}([\mathbf{x}_e^t, \mathbf{M}', \mathbf{x}_s], t, P_{\text{final}})$
7:     $\epsilon_{\text{guided}} \leftarrow \epsilon_{\text{uncond}} + \mathbf{M}_{\text{cfg}} \cdot (\epsilon_{\text{cond}} - \epsilon_{\text{uncond}})$
8:     $\mathbf{x}_e^{t-1} \leftarrow S.\text{step}(\mathbf{x}_e^t, \epsilon_{\text{guided}}, t)$
9: **end for**
10: $\hat{\mathbf{x}}_0 \leftarrow \text{DecodeLatent}(\mathbf{x}_0)$
11: **return** $\hat{\mathbf{x}}_0$

---

and temporally. MASG applies distinct guidance scales to masked and unmasked regions throughout the denoising process.

The design of MASG is motivated by the operational dynamics of the diffusion denoising process. Seminal work Yi et al. (2024) has established that the model synthesizes low-frequency information, such as coarse structures and shapes, during the initial denoising timesteps. Conversely, high-frequency details, like textures and fine-grained features, are rendered in the final stages. To leverage this phenomenon for seamless inpainting, our scheduling strategy enforces strong guidance early on to generate a coherent structure within the masked region, and then gradually reduces it to facilitate the generation of consistent, high-fidelity textures that blend harmoniously with the surrounding background.

Formally, at each reverse-diffusion timestep $t$ (from $T$ down to 1), we construct a pixel-wise guidance map $\mathbf{M}_t^{\text{cfg}} \in \mathbb{R}^{h \times w}$, where $h \times w$ are the dimensions of the latent representation. This map is derived from the down-scaled binary mask $\mathbf{M}' \in \{0, 1\}^{h \times w}$:

$$\mathbf{M}_t^{\text{cfg}}(i, j) = \begin{cases} \gamma_t^{\text{mask}}, & \mathbf{M}'(i, j) = 1 \text{ (masked region)}, \\ \gamma^{\text{bg}}, & \mathbf{M}'(i, j) = 0 \text{ (background)}. \end{cases} \tag{2}$$

Here, $\gamma^{\text{bg}}$ is a constant guidance scale for the background, while $\gamma_t^{\text{mask}}$ is the scheduled guidance for the inpainted region. The schedule maintains a high, constant guidance, $\gamma_T^{\text{mask}}$, from the initial timestep $T$ until a specified cutoff timestep $t_c$. For subsequent steps ($t \leq t_c$), it smoothly interpolates towards the background guidance $\gamma^{\text{bg}}$ using a cosine schedule. This is defined as:

$$\gamma_t^{\text{mask}} = \begin{cases} \gamma_T^{\text{mask}}, & t_c \leq t \leq T, \\ \gamma^{\text{bg}} + s(t) \left( \gamma_T^{\text{mask}} - \gamma^{\text{bg}} \right), & 1 \leq t < t_c, \end{cases} \tag{3}$$

where $s(t)$ is the cosine decay scheduler and is defined as:

$$s(t) = \frac{1 - \cos\left(\pi \frac{t-1}{t_c - 1}\right)}{2}. \tag{4}$$

Algorithm 1 details the full pipeline. Furthermore, the different scheduling options are examined in Appendix C.

## 4 EXPERIMENTS

### 4.1 IMPLEMENTATION DETAILS

**Training and Testing Datasets.** For fine-tuning, we employed the RoRem dataset Li et al. (2025) alongside our proposed STORe dataset to adapt the SDXL Inpainting model Podell et al. (2023). For evaluation, we utilized two distinct benchmarks. To assess standard fidelity metrics, we followed the protocol from Li et al. (2025) and used a test set of 10,000 images randomly selected from

Table 1: Quantitative comparison of object removal methods at resolution of $1024 \times 1024$ on the LRNR test set for the Success Rate metric and the OpenImages-10k benchmark. The best-performing model for each metric is highlighted in bold, and the second-best is shown in underscore.

| Method | Success Rate↑ | FID↓ | Local FID↓ | PSNR↑ | LPIPS↓ | CLIP Score↑ |
|---|---|---|---|---|---|---|
| MAT (CVPR '22) Li et al. (2022a) | 20.5 | 2.38 | 5.89 | 22.09 | 0.22 | 22.16 |
| Lama (WACV '22) (Big) Suvorov et al. (2021) | 21.1 | 1.98 | 10.49 | 22.03 | **0.17** | **23.77** |
| LDM (CVPR '22) Rombach et al. (2021a) | 15.6 | 1.82 | 8.73 | **22.43** | 0.24 | 23.16 |
| PowerPaintV2 (ECCV '24) Zhuang et al. (2024) | 16.2 | 1.83 | 5.13 | 19.78 | 0.30 | 22.22 |
| Attentive Eraser (AAAI '25) Sun et al. (2025) | 30.7 | 2.74 | 8.54 | 19.99 | 0.31 | 22.88 |
| CLIPAway (NeurIPS '24)Ekin et al. (2024) | 7.9 | 2.69 | 27.62 | 18.78 | 0.40 | 21.70 |
| RoRem-Mixed (CVPR '25) Li et al. (2025) | 60.7 | 1.92 | 8.61 | 21.14 | 0.29 | 22.84 |
| SDXL Inpainting Podell et al. (2023) | 5.5 | **0.84** | **3.56** | 22.17 | 0.26 | 22.21 |
| **LRNR** | **76.4** | 1.30 | 7.09 | 21.06 | 0.26 | 23.03 |

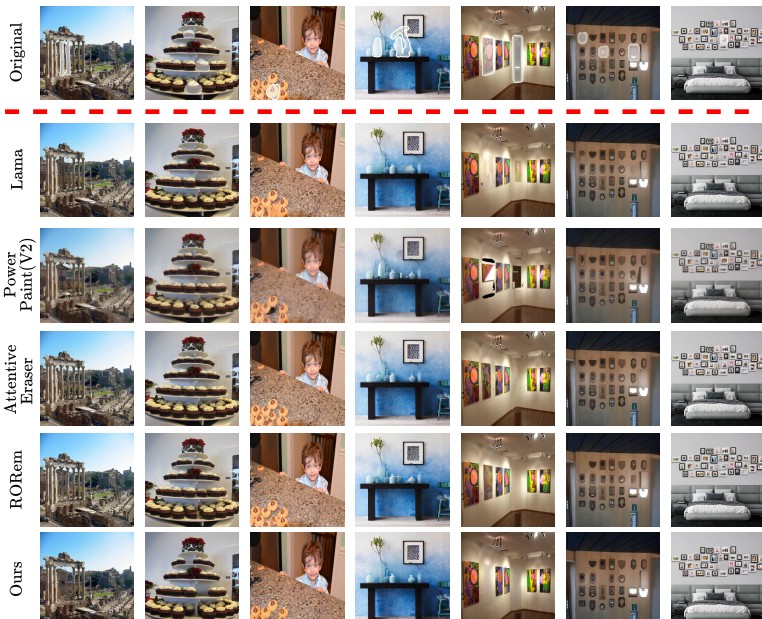

Figure 4: **Qualitative comparison on challenging object removal scenarios.** From top to bottom: Original image, Lama, PowerPaint(V2), Attentive Eraser, RoRem, and our model. Competing methods often produce undesirable artifacts, including blurry reconstructions (Lama) or the replication of surrounding objects (Attentive Eraser, RoRem). Our model consistently generates clean and plausible results without these failure modes.

OpenImages-V6, excluding samples with masks covering less than 3% or more than 70% of the image. To specifically evaluate performance in difficult scenarios, we curated the **LRNR test set**, a new benchmark of 200 images featuring challenging scenes with cluttered and repetitive objects. This specialized test set was used to measure the human-perceived Success Rate.

**Model Training.** The Stable Diffusion XL Inpainting model Podell et al. (2023) was used as our base model. As illustrated in Figure 2, the SDXL UNet and Text Encoder parameters was kept frozen, with training applied exclusively to the LoRA modules and the embedding vectors associated with $P_{rmv}$. In total, this corresponds to fine-tuning **less than 2%** of the SDXL parameters (see Appendix G for more details). We employed the AdamW optimizer Loshchilov & Hutter (2017) with distinct learning rates: 5e-4 for the LoRA parameters and 2.5e-4 for the embedding vectors of $P_{rmv}$. We applied LoRA with a rank of $r = 32$ to the query, key, and value projections in both the self-attention and cross-attention layers of the UNet. Each training run consisted of 1k iterations on two NVIDIA RTX 3090 GPUs. For the text encoder, the single-token prompt $P_{rmv}$ was initialized with the pre-trained embedding vector for the word "Remove". This initial embedding was subsequently fine-tuned during training. At inference time, the fixed prompt $P_{rmv}$ is prepended to each input instance to guide the object removal process.

Table 2: Ablation study of our model's components at resolution of $1024 \times 1024$ on the OpenImages-10k and the LRNR test datasets.

| Training Dataset | Model | Success Rate↑ | FID↓ | Local FID↓ | PSNR↑ | LPIPS↓ | CLIP Score↑ |
|---|---|---|---|---|---|---|---|
| RoRem + STORe | SDXL Inp. + LORA (Baseline) | 10.1 | **0.62** | **3.02** | **22.74** | **0.22** | 22.38 |
| | Baseline + $P_{\text{rmv}}$ (77T) | 30.2 | 0.87 | 4.65 | 21.47 | 0.24 | 23.07 |
| | Baseline + $P_{\text{rmv}}$ (1T) | 70.8 | 6.10 | 8.40 | 20.97 | 0.50 | **24.24** |
| | Baseline + $P_{\text{rmv}}$ (1T) + $P_{\text{enh}}$ | 74.2 | 1.60 | 6.99 | 20.72 | 0.33 | 23.07 |
| | Baseline + $P_{\text{rmv}}$ (1T) + $P_{\text{enh}}$ + MASG | **76.4** | 1.30 | 7.09 | 21.06 | 0.26 | 23.03 |
| RoRem | Baseline + $P_{\text{rmv}}$ (1T) + $P_{\text{enh}}$ + MASG | 70.3 | 1.41 | 7.84 | 20.97 | 0.26 | 23.09 |

**Baselines.** We benchmark our method against multiple state-of-the-art (SOTA) object-removal approaches, with detailed baseline descriptions provided in Appendix B.

**Evaluation Metrics.** We evaluated our method using a combination of standard automated metrics and a formal user study. To assess image quality, we used PSNR, FID, and LPIPS for global fidelity, and Local-FID Xie et al. (2023a) to specifically evaluate the quality of the inpainted region, following established practices Li et al. (2025); Suvorov et al. (2021); Zhuang et al. (2023). To measure how effectively objects were removed, we computed the CLIP score Radford et al. (2021); Lu et al. (2024); Liu et al. (2024) between the generated patch and the prompt "background," where a higher value signifies a more complete removal. Since these metrics may not fully capture human perception, we also conducted a user study involving eight volunteers. Participants evaluated each output as either a success or failure, based on the completeness of the object removal and the visual quality of the inpainted area. Further details of the study are provided in Appendix A.

## 4.2 OBJECT REMOVAL RESULTS

**Quantitative Comparison.** Our quantitative evaluation, presented in Table 1, benchmarks LRNR against state-of-the-art methods on two fronts: task-specific performance via Success Rate on our challenging LRNR test set, and image fidelity on the OpenImages-10k benchmark. The results demonstrate that LRNR achieves a superior balance between these two objectives. The most critical metric is the human-judged Success Rate, where LRNR establishes a new state-of-the-art at **76.4%**. This significantly outperforms all competitors, including the next-best specialized model, RoRem-Mixed (60.7%), underscoring our method's advanced ability to correctly interpret and execute the removal task.

Beyond task success, LRNR also excels in image fidelity. It achieves a strong FID score of **1.30**, notably better than other task-focused methods like RoRem-Mixed (1.92). While the general-purpose SDXL Inpainting model has a lower FID (0.84), its negligible Success Rate (5.5%) makes it unreliable for this task. LRNR, therefore, presents the most effective trade-off between successful removal and photorealism. This balance is further supported by a competitive LPIPS score of 0.26, matching the strong SDXL baseline. Crucially, as detailed in Appendix G, this superior performance is achieved with remarkable efficiency. Even when trained on the same dataset as RoRem-Mixed, LRNR yields better results across Success Rate, FID, LPIPS, and CLIP Score while training approximately **71x fewer parameters**. This demonstrates the profound efficiency of our parameter-efficient strategy and MASG pipeline compared to traditional full fine-tuning.

**Qualitative Comparison** Figure 4 qualitatively compares our model against leading methods on object removal tasks with repetitive surrounding textures. The selected images are specifically chosen to highlight a common challenge in object removal: removing a target that is surrounded by repetitive textures or structurally similar objects. We test both **Hard** cases, where the object is occluded with little background context (e.g., child and cookies, column 3), and **Normal** cases with more background information (e.g., gallery painting, column 5). As shown, competing models often produce blurry artifacts (LaMa, PowerPaint(V2)), incomplete removal, or erroneously replicate surrounding textures and objects (Attentive Eraser, RORem). In contrast, our model excels in both scenarios, generating clean, artifact-free results without object replication. The model's robustness to cross-domain generalization is further evaluated in Appendix H.

## 4.3 ABLATION STUDY

**LRNR Components.** Our systematic ablation study, summarized in Table 2, validates each component of LRNR by incrementally building upon an `SDXL Inp.+ LORA` baseline. First, intro-

Table 3: A Quantitative study of PowerPaint performance through the addition of the MASG: Analysis of various modes of guidance scale on the LRNR test dataset.

| Method | Guidance scale | Success Rate↑ | PSNR↑ | LPIPS↓ | CLIP Score↑ |
|---|---|---|---|---|---|
| PowerPaint (V2) | 12 | 16.2 | 17.49 | 0.38 | 22.31 |
| PowerPaint (V2) w/ MASG | 20 → 1 | 27.3 | 18.19 | 0.34 | 22.29 |
| LRNR-1T | 7.5 | 72.6 | 19.15 | 0.35 | 22.70 |
| LRNR-1T w/ MASG | 12.5 → 1 | 76.4 | 20.30 | 0.22 | 22.37 |

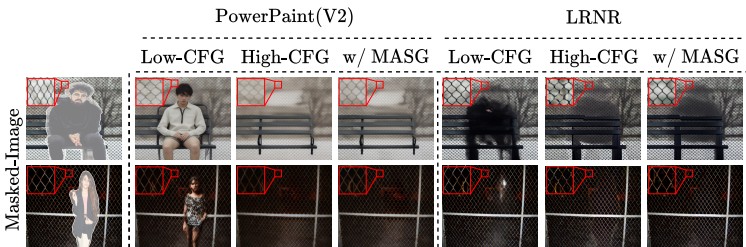

Figure 5: Visual comparison of object removal and pattern reconstruction in PowerPaint and our LRNR model at varying guidance scales (GS), with MASG enabled.

ducing a single-token learnable removal prompt ($P_{\text{rmv}}$) dramatically boosted the Success Rate from 10.1% to 70.8%, though it severely degraded image fidelity (FID worsened from 0.87 to 6.10). To counteract this, we introduced a static enhancement prompt ($P_{\text{enh}}$) in a dual-prompt strategy, which successfully restored a competitive FID of 1.60 while further increasing the Success Rate to 74.2%, effectively decoupling the removal and generation tasks. Next, applying our training-free MASG technique at inference pushed performance to its peak, achieving a 76.4% Success Rate and a 1.30 FID. Finally, training our full model on the RoRem dataset instead of ours caused the Success Rate to drop to 70.3%, confirming that our STORe dataset is crucial for achieving state-of-the-art robustness. Each component thus provides a distinct, synergistic benefit, culminating in a model that balances removal effectiveness and photorealism, with further qualitative results in Appendix F.

**MASG Ablation.** To validate the effectiveness and generality of our Mask-Aware Scheduled Guidance (MASG), we applied it to other task-prompt-based models, using PowerPaint Zhuang et al. (2023) as a representative baseline. The goal was to test if MASG could mitigate the common trade-off where a high guidance scale (GS) improves task adherence but degrades image quality with artifacts like color bleeding and texture inconsistency.

The results, presented in Table 3, demonstrate that MASG is highly effective. When integrated into PowerPaint, it yields significant improvements across all fidelity metrics, achieving higher PSNR and lower LPIPS and FID scores. This shows that MASG successfully enforces prompt fidelity while preserving high visual quality. The qualitative results in Figure 5 further illustrate these dynamics: a low GS leads to incomplete removal, a high GS causes artifacts, but MASG produces results that are both correctly removed and seamlessly blended.

By transitioning the guidance from high to low as defined in Equation 3, MASG allows the model to first establish the correct structure for removal and then render high-frequency details that harmonize with the background. This confirms that MASG is a general, training-free technique that robustly balances task compliance and photorealism in guided inpainting models.

## 5    CONCLUSION

We introduced the LRNR, a framework that solves object replication in diffusion-based removal. Our method combines the specialized STORe dataset for robustness, a learnable prompt ($P_{rmv}$) , and MASG for visual quality. LRNR achieves a state-of-the-art 76.4% success rate, significantly outperforming existing methods while maintaining high image fidelity. Ablation studies confirm each component is crucial, establishing LRNR as a significant advancement in reliable image editing and object removal task.

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

# APPENDIX

## A   USER STUDY

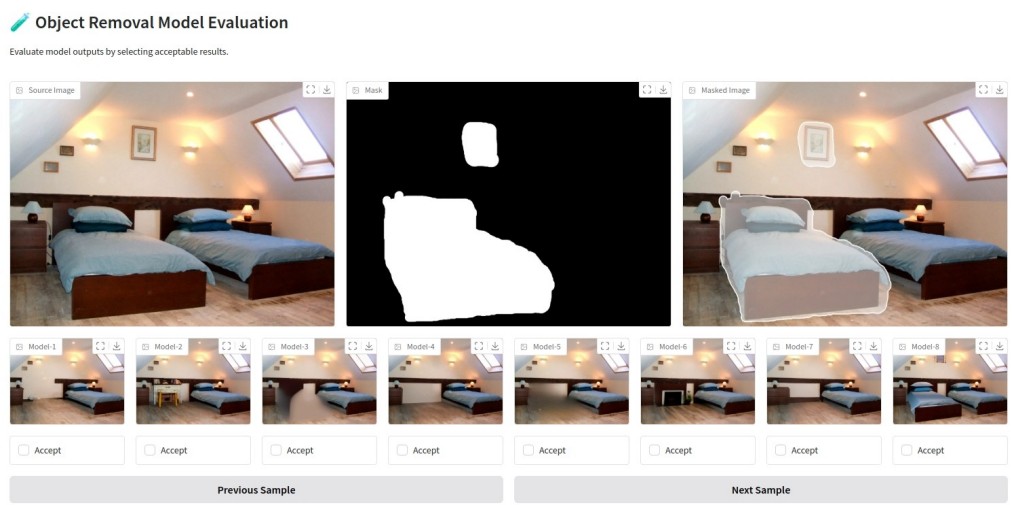

Figure 6: The human evaluation interface. Volunteers independently marked a result as a 'Success' by selecting the 'Accept' checkbox.

To ensure a fair evaluation of our model's performance from a human-centric perspective, we conducted a user study. This section details the protocol and participant instructions used to assess the "Success Rate" metric reported in section 4.1 of the main paper.

### A.1   STUDY DESIGN AND PARTICIPANTS

The user study was designed to gather subjective feedback on the quality of object removal from various models.

- **Participants:** The study involved eight volunteers. The participants were graduate students in computer science with a background in computer vision and image processing. This selection ensures that the evaluators possessed a sufficient level of expertise to critically assess image quality and identify subtle artifacts, but were not involved in the development of this project to avoid bias.

- **Anonymization:** All results presented to the participants were anonymized. The outputs from our model (LRNR) and the baseline models were shuffled randomly. Participants were not aware of which model generated which image, preventing any potential bias towards our proposed method.

- **Evaluation Interface:** Participants used a simple web interface that displayed the original image, mask, and masked image indicating the object for removal. They were then shown the outputs from all models and asked to independently assess each result as a 'Success' or 'Failure'. We developed this evaluation interface as a web application using the open-source Gradio library Abid et al. (2019). The structure of the interface is illustrated in Figure 6.

### A.2   INSTRUCTIONS FOR PARTICIPANTS

To ensure consistency across evaluations, all participants were provided with the following set of instructions:

"You will be shown a series of images. For each example, you will see the original image with a masked object and the result after an object removal algorithm has

been applied. Your task is to evaluate whether the object removal was a **'Success'** or a **'Failure'**.

An output should be marked as a **'Success'** only if it meets both of the following criteria:

1. **Complete and Clean Removal:** The target object specified by the mask must be entirely gone. There should be no residual artifacts, "ghosts," or fragments of the original object remaining in the inpainted region.

2. **Plausible and High-Quality Inpainting:** The filled area must be semantically coherent and blend seamlessly with the surrounding background. The generated content should be plausible, realistic, and free of visual distortions, such as blurriness, unnatural textures, or color inconsistencies.

The "Success Rate" is finally calculated as the mean percentage of outputs rated as 'Success' by the participants for each model.

## B  BASELINES

We compare our method against several state-of-the-art (SOTA) object removal baselines. Our selection criterion was the public availability of source code and models. The chosen methods include LaMa Suvorov et al. (2021), LDM Rombach et al. (2021a), SDXL-Inpainting Podell et al. (2023), CLIPAway Ekin et al. (2024), PowerPaint Zhuang et al. (2023), AttentiveEraser Sun et al. (2025), and RoRem Li et al. (2025). Consequently, we excluded methods without public implementations, such as ObjectDrop Winter et al. (2024) and MagicEraser Li et al. (2024). For PowerPaint, RoRem and LaMa, we utilized their enhanced and publicly released versions—PowerPaint (V2), RoRem-Mixed and LaMa (Big)—which are superior to the models originally presented in their respective papers.

## C  ABLATION ON MASG SCHEDULING

We conducted an ablation study to determine the optimal scheduling strategy for our proposed Mask-Aware Scheduled Guidance (MASG).We evaluated several scheduling functions that transition the scale from an initial high value, $\gamma_T^{\text{mask}}$, to a final low value, $\gamma^{\text{bg}}$. The primary schedulers tested include:

- **Linear:** Decreases the guidance scale at a constant rate.

$$\gamma_t^{\text{mask}} = \gamma_T^{\text{mask}} - (\gamma_T^{\text{mask}} - \gamma^{\text{bg}}) \times \frac{T - t}{T - 1}$$

- **Cosine:** Employs a smooth cosine annealing curve for a more gradual transition.

$$\gamma_t^{\text{mask}} = \gamma^{\text{bg}} + \frac{1}{2}(\gamma_T^{\text{mask}} - \gamma^{\text{bg}})\left(1 + \cos\left(\frac{T - t}{T - 1}\pi\right)\right)$$

- **One-step:** Maintains high guidance until a specific timestep $T_{\text{step}}$, then abruptly drops to the minimum.

$$\gamma(t) = \begin{cases} \gamma_T^{\text{mask}} & \text{if } t_c \leq t \leq T \\ \gamma^{\text{bg}} & \text{if } 1 \leq t < t_c \end{cases}$$

In addition to these, we evaluated two hybrid methods, `Linear + Step` and `Cosine + Step`. The quantitative results, presented in Table 4, demonstrate that the choice of scheduler has a significant impact on final image quality. Notably, the `Cosine` scheduler excels in reconstruction-oriented metrics, achieving the highest PSNR (21.28) and the lowest Local-FID (6.74). This suggests that a smooth, cosine-based decay is highly effective at ensuring the inpainted region is locally coherent and accurate.

Upon further analysis, the hybrid `Cosine + Step` scheduler emerged as the most robust choice overall. It achieved the best FID score of 1.30, indicating superior global image realism compared

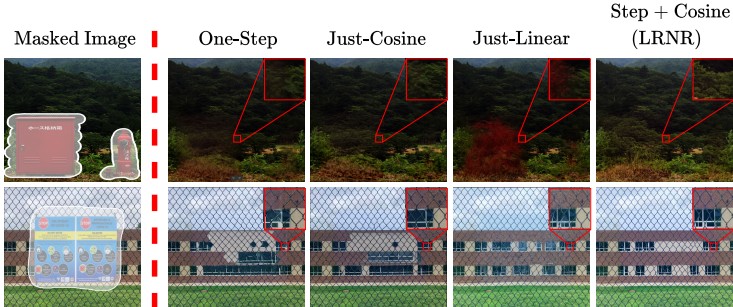

Figure 7: Qualitative analysis of MASG scheduling strategies. While simpler schedulers like `One-Step` and `Just-Cosine` trade off removal effectiveness for image quality, our final method, `Step + Cosine`, demonstrates superior performance by achieving both complete object erasure and high-fidelity texture reconstruction, as shown in the magnified regions.

Table 4: Ablation study on different scheduling strategies for Mask-Aware Scheduled Guidance (MASG). All evaluations are performed on a OpenImages-10K image dataset at 1024×1024 resolution. The best result for each metric is shown in bold.

| Scheduler | FID ↓ | Local-FID ↓ | PSNR ↑ | LPIPS ↓ | CLIP Score ↑ |
|---|---|---|---|---|---|
| Linear | 1.40 | 7.34 | 20.82 | **0.26** | 23.04 |
| Cosine | 1.43 | **6.74** | **21.28** | 0.27 | 22.96 |
| One-step | 1.35 | 7.27 | 20.64 | **0.26** | **23.05** |
| Cosine + Step | **1.30** | 7.09 | 21.06 | **0.26** | 23.03 |
| Linear + Step | 1.34 | 7.15 | 20.59 | **0.26** | 23.04 |

to all other variants. While the pure `Cosine` schedule showed a slight advantage in local metrics, the `Cosine + Step` strategy strikes a more effective balance between preserving background integrity, ensuring global coherence, and achieving high-fidelity inpainting. Given that its leading FID score signifies a generated distribution closer to that of real images, we adopt the `Cosine + Step` scheduler for MASG in all our main experiments to ensure the most plausible and artifact-free results.

Figure 7 provides a qualitative comparison of the different scheduling strategies. The `Linear` scheduler demonstrates weak performance, often resulting in incomplete object removal and noticeable boundary artifacts around the inpainted region. While the `One-step` scheduler achieves more effective erasure, the generated pixels suffer from blurring and a lack of fine-grained texture. Conversely, the `Cosine` scheduler produces higher-quality textures than both `Linear` and `One-step`, but its removal efficacy is inferior to the `One-step` approach. Consequently, the hybrid `Cosine + Step` strategy yields the best visual results, successfully balancing complete object removal with the generation of high-fidelity, artifact-free content. Based on these superior qualitative and quantitative results, we adopt `Cosine + Step` as our scheduling technique for all experiments.

# D    ADDITIONAL QUALITATIVE RESULTS

Figure 9 presents a qualitative comparison of our proposed method with the remaining inpainting models from Table 1: Clipaway, SDXL-INP, and LDM, on the challenging task of object removal. The top row displays the original images, while the subsequent rows show the results after removing a masked object from each scene. Baseline models like LDM and SDXL-INP occasionally introduce noticeable artifacts, such as blurry textures or incomplete removal. Clipaway struggles with cluttered scenes, as seen in the cupcake image (second column), and with generating coherent content in the masked region of the image with the boy (third column). In contrast, our method consistently generates more photorealistic and coherent results across a diverse range of images. It successfully reconstructs intricate details, such as the cupcake stand's structure and the complex arrangement of picture frames, while maintaining global semantic consistency and producing visually appealing outputs that are often indistinguishable from the original background.

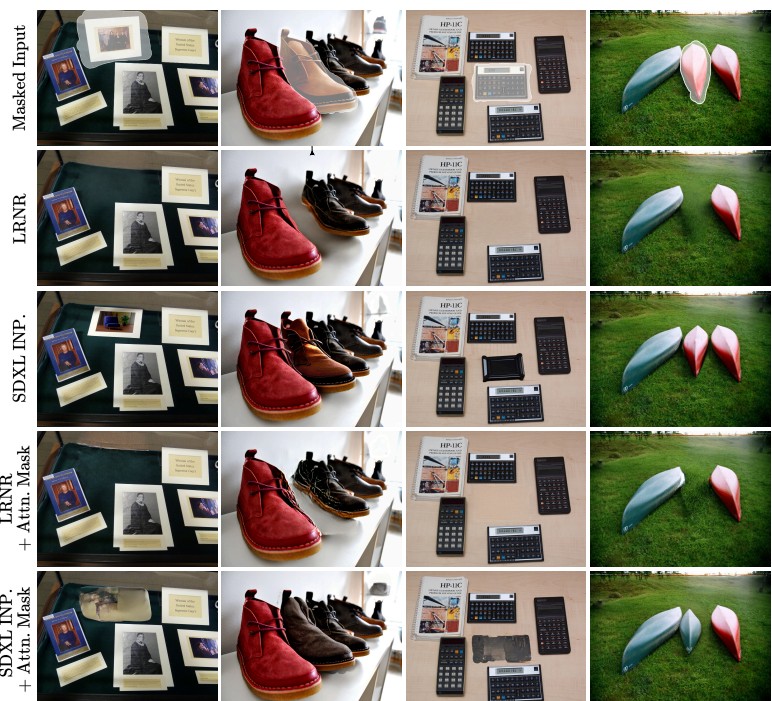

Figure 8: Qualitative comparison of our proposed LRNR model against baselines using an attention mask. Applying the mask to SDXL Inpainting improves removal but remains inferior to our method. Applying the same mask to our LRNR model degrades its performance, demonstrating the superiority of its learned contextual reasoning.

## E    COMPARISON WITH ATTENTION MASK METHOD

To evaluate our model against training-free methods that utilize attention manipulation, we conduct a comparative analysis. We first compare our LRNR model's performance against a strong baseline: the SDXL Inpainting model augmented with a self-attention masking technique. This approach is representative of methods that manipulate the UNet's attention mechanism to improve object removal, as explored in prior work Li et al. (2024); Sun et al. (2025). For this baseline, we adapt the self-attention masking strategy from EraseDiffusion Liu et al. (2025). Furthermore, to test our hypothesis about the limitations of such explicit masking, we extend this experiment by applying the identical attention mask logic to our own LRNR model.

In the attention-masking setup, for each self-attention layer $l$ operating on a feature map of spatial dimensions $h \times w$, we dynamically construct a corresponding attention mask. The high-resolution binary input mask $\mathbf{M} \in \mathbb{R}^{H \times W}$ is first down-sampled using interpolation to match the layer's feature resolution, yielding $\mathbf{M}_l \in \mathbb{R}^{h \times w}$. This layer-specific mask is then flattened into a vector containing $N = h \times w$ tokens. From this, we construct the final attention mask $M_l^{attn} \in \mathbb{R}^{N \times N}$, which controls the information flow between query-key token pairs. This relationship can be expressed as:

$$M_l^{attn}[:,j] = \left\{ \begin{array}{ll} 1, & j \in F_l^{obj} \\ 0, & j \in F_l^{bg} \end{array} \right. \tag{5}$$

where $F_l^{obj}$ is the set of token indices corresponding to the masked object and $F_l^{bg}$ is the complementary set of background tokens.

As illustrated in Figure 8, applying the self-attention mask to the standard SDXL Inpainting model enhances its object removal capability, yet its performance remains inferior to our LRNR model. The SDXL + Attn. Mask baseline still exhibits critical failure modes, including incomplete generation, remnant artifacts (ghosting), and at times, regeneration of the removed object.

Table 5: Our LRNR model significantly outperforms the fully fine-tuned RoRem-Mixed, achieving a higher Success Rate and superior image fidelity. This result, obtained while training on the same core dataset and using fewer parameters, highlights the power of our efficient pipeline, which combines LoRA, Textual Inversion, and our MASG inference strategy.

| Method | Success Rate↑ | FID↓ | Local FID↓ | PSNR↑ | LPIPS↓ | CLIP Score↑ |
|---|---|---|---|---|---|---|
| RoRem-Mixed Li et al. (2025) | 60.7 | 1.92 | 8.61 | **21.14** | 0.29 | 22.84 |
| LRNR w/o STORe | **70.3** | **1.41** | **7.84** | 20.97 | **0.26** | **23.09** |

Notably, when we apply the same attention masking technique to our proposed LRNR model, we observe not an improvement, but a degradation in image quality. As seen in the figure, the output from LRNR + Attn. Mask is of lower quality than that of the standalone LRNR model, demonstrating that forcing such a constraint on our model is counterproductive. We believe that the issues with attention masking stem from its brute-force nature. By strictly forbidding attention to the object region, the model is prevented from utilizing potentially useful contextual cues—like the shape, lighting, and texture immediately surrounding the mask—that are critical for seamless blending.

## F  ANALYSIS OF MODEL COMPONENTS

To further illustrate the contribution of each component in our model, Figure 10 presents qualitative examples of the progressive improvements in both object removal and output quality across successive training stages. Starting with the SDXL + LoRA baseline, we observe that while the inpainting is visually coherent, the model fails to fully remove the object, leaving behind residual structures or semantically similar content. When incorporating the 77-token, the outputs appear slightly more aligned with the removal intent, but performance remains inconsistent. The most notable change occurs after switching to a focused 1-token prompt, where the model learns a precise representation of the object removal task. At this stage, the model exhibits significant improvements in object elimination, although with a degradation in quality (e.g., blurriness or unnatural textures). To address this, we introduce $P_{enh}$ to preserve texture quality and scene realism. As shown in the figure, this addition restores photorealism while maintaining the semantic clarity of the removal task. Finally, by applying our Mask-Aware Scheduled Guidance (MASG) at inference time, we observe further improvements in both removal completeness and blending quality. MASG reduces residual artifacts, enhances structural consistency, and ensures that the inpainted region remains visually seamless with the unmasked context without requiring additional training.

## G  MODEL EFFICIENCY

The core of LRNR's design is a pipeline that prioritizes both computational efficiency and generative quality. Unlike methods such as RoRem that rely on resource-intensive full fine-tuning, our approach is holistically superior. We pair an efficient training strategy to reduce trainable parameters by nearly 71x with a novel, training-free inference technique, MASG. The efficient training teaches the model the concept of removal, while MASG intelligently guides the application of this concept during generation. Additionally, our training process updates only ∼35 million parameters, representing just a few training parameters (less than 2%) compared to full fine-tuning. This leads to faster training and a final model checkpoint of only a few MB, compared to the ∼5 GB checkpoint of a fully fine-tuned model. As our results demonstrate in Tabe 5, this complete system not only drastically reduces computational overhead but also achieves a higher success rate and superior image fidelity, proving that a more intelligent, parameter-efficient framework is more effective than a brute-force one.

## H  CROSS-DOMAIN GENERALIZATION

To evaluate the generalization ability of object removal models across diverse image styles, Figure 11 presents representative examples from five distinct style categories: cartoon, comic, LEGO, anime, and neon. Some of these samples were generated using diffusion models, which introduce challenging conditions for object removal. In the first comic-style sample, none of the competing methods manage to remove the window properly. Most either replace it with a distorted version or

leave visible remnants. In contrast, our model cleanly removes the window and restores a visually coherent wall. In the first LEGO-style image, while most competing methods succeed in removing the object, they leave behind visual artifacts such as disrupted structural patterns. Our model, by contrast, achieves a plausible restoration that preserves these patterns. In the anime-style example, methods like RoRem fail to generate coherent facial structures. In contrast, our model generates smooth, stylistically consistent faces that blend seamlessly with the original anime composition. Our model, through a combination of prompt-aware training and Mask-Aware Scheduled Guidance, adapts more effectively to these varied visual domains and delivers perceptually clean removals.

## I  QUALITY-PRESERVING PROMPT

During training, we construct a quality-preserving prompt $P_{\text{enh}}$ by randomly selecting some of several high-fidelity descriptor tokens. These tokens encourage improvements in resolution, detail, and overall realism. In the following, some example prompt that is used for the $P_{enh}$ is provided:

- `high-quality`
- `best-texture`
- `ultra-detailed`
- `Full-HD`
- `HDR`
- `full dynamic range`
- `photorealistic`
- `8K`
- `4K`
- `nikon`
- `best-result`
- `high-resolution`
- `awesome-detail`

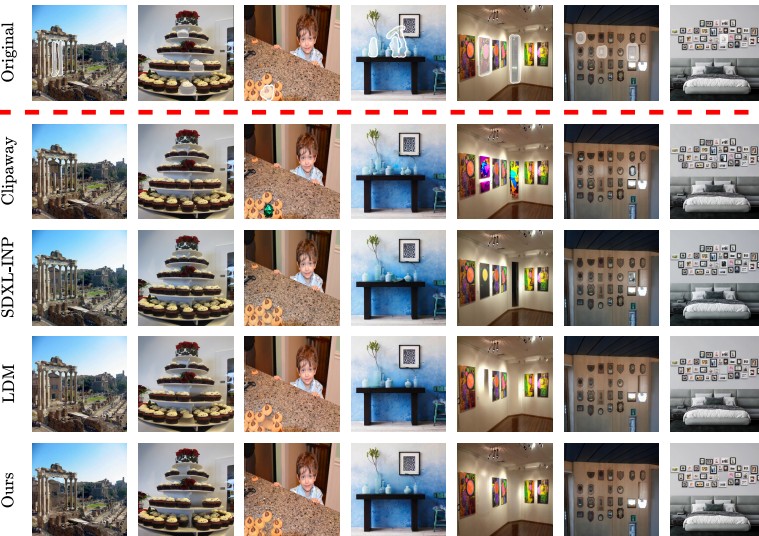

Figure 9: Qualitative comparison of object removal performance across methods. From top to bottom: original image, CLIP-Away, SDXL, LDM, and LRNR (ours).

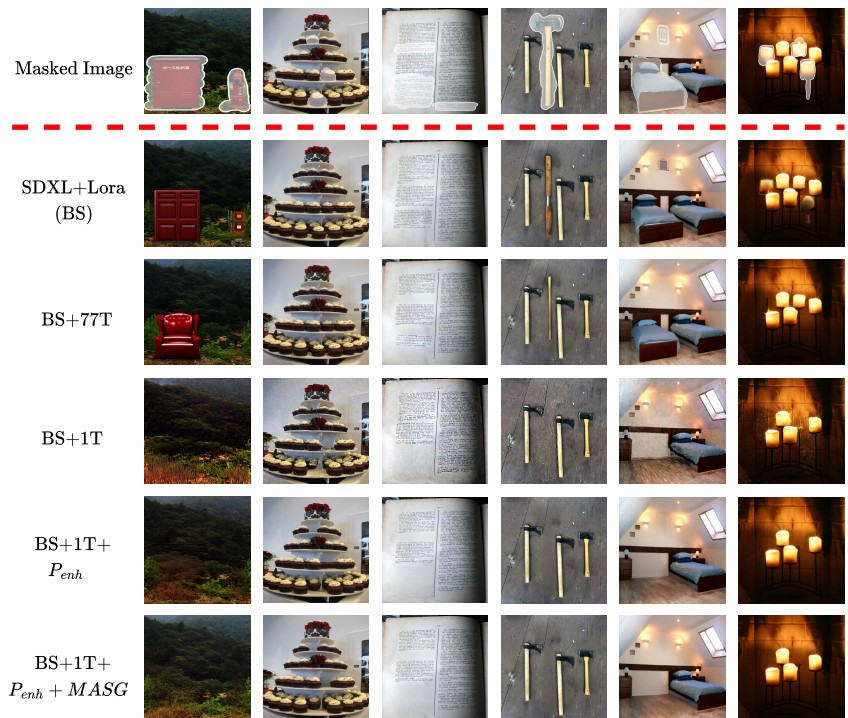

Figure 10: Qualitative progression of object removal and image quality through different stages of model development. Improvements in removal accuracy and texture quality become increasingly visible through each step.

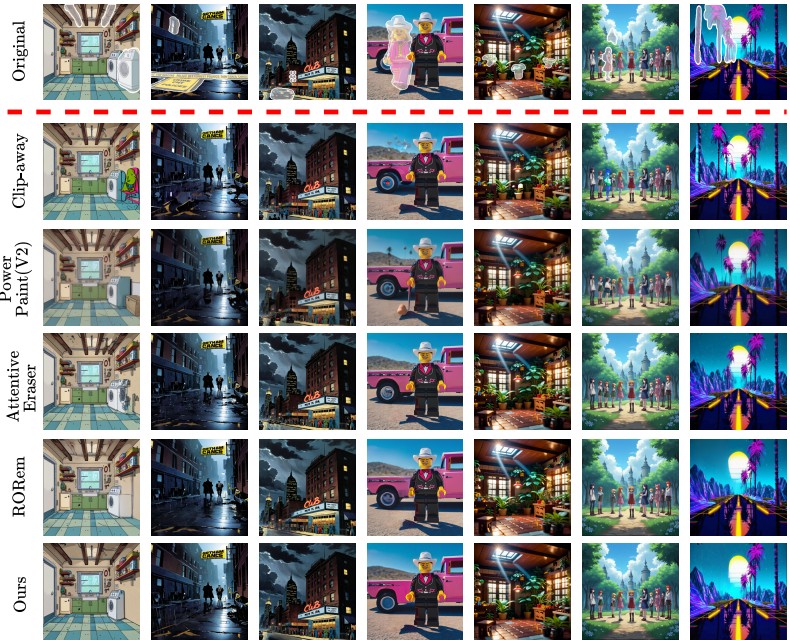

Figure 11: Qualitative comparison of object removal performance across diverse visual domains, including cartoon, comic, LEGO, anime, and neon style scenes. Our method consistently achieves complete removal with clean reconstructions, while other models struggle with artifacting, incomplete deletion, or object regeneration.

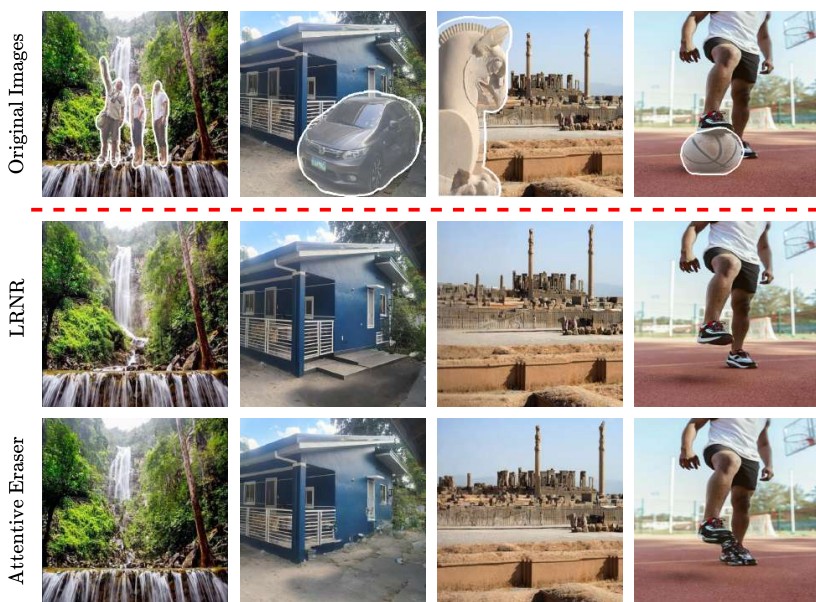

Figure 12: Generalization to Non-Cluttered Scenes. Qualitative comparison between our proposed LRNR (second row) and Attentive Eraser (third row) on standard, non-cluttered object removal tasks. The columns display diverse scenarios: (1) natural landscape, (2) residential structure, (3) historical ruins, and (4) dynamic action. The results demonstrate that LRNR effectively generalizes to simple inputs without suffering from catastrophic forgetting, producing high-fidelity inpainting results comparable to specialized baselines even when the complex "clutter" correlations are absent.

## J EVALUATION ON NON-CLUTTERED SCENES

To assess the risk of catastrophic forgetting, we evaluate LRNR on **default non-cluttered scenes** (Figure 12) in comparison to the Attentive Eraser baseline. The qualitative results demonstrate that our model retains robust general inpainting capabilities: LRNR successfully removes isolated foreground objects—such as pedestrians in a natural landscape (Column 1) or a vehicle in a residential driveway (Column 2)—while preserving the structural integrity of the background. As observed in the waterfall and ruins examples (Columns 1 & 3), our method synthesizes plausible textures (e.g., continuous water flow and stone masonry) that are semantically consistent with the surrounding context, matching the visual fidelity of the baseline. Crucially, this confirms that fine-tuning on the specialized STORe dataset effectively instills the remove concept without degrading the model's ability to handle simple, sparse scenarios or introducing unwanted artifacts in clean backgrounds.

## K LIMITATIONS

Figure 13 illustrates some of the limitations of our LRNR method. These issues stem primarily from the generative priors of the underlying SDXL backbone. One such issue is Semantic Persistence (Column 1), where strong semantic associations override the removal intent; for example, when masking a license plate, the model generates a new code rather than erasing it because its internal world model dictates that cars must have plates. Another limitation is Contextual Hallucination (Column 2), which occurs when the surrounding context strongly implies an object's presence; here, the man's grasping hand pose forces the model to generate gloves to make the geometry plausible. Additionally, we observe that the model struggles with facial reconstruction in these complex scenarios, leading to visible artifacts on the subject's face.

Finally, we encounter Large Mask Degradation (Column 3) when removing larger objects like the central truffle. As the mask size increases, we observe a drop in the quality of the generated pixels, resulting in blurriness compared to the sharp surrounding texture. We believe these quality issues are largely architectural; they can likely be resolved in future iterations by transitioning our pipeline

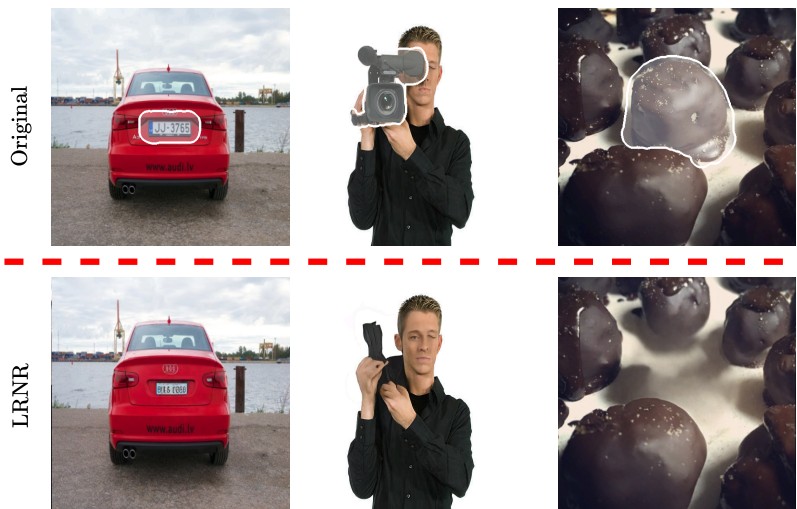

Figure 13: Analysis of removal failures. These examples highlight the tension between removal intent and generative priors. Column 1 shows the model enforcing semantic logic (cars have plates); Column 2 shows physical plausibility overriding the mask (hands must hold something); and Column 3 illustrates texture quality degradation in large masked regions..

to stronger, next-generation baselines like FLUX Labs (2024), which demonstrate improved high-frequency texture generation and facial consistency.

## L    MASK SENSITIVITY

Here, we conducted a systematic ablation study evaluating the impact of mask dilation (over-masking) and erosion (under-masking). We applied varying degrees of random dilation and erosion (ranging from $0\%$ to $30\%$) to the input masks during inference to simulate different levels of user error. The results, illustrated in Figures 14 and 15, reveal a clear asymmetry in the model's robustness.

**Robustness to Dilation (Over-Masking).**    As detailed in Figure 14, our method demonstrates high robustness to mask expansion. The Success Rate remains stable between $D = 5\%$ (77.5%) and $D = 10\%$ (77.2%), indicating that users do not need pixel-perfect masks for effective removal. However, as dilation exceeds $15\%$, we observe a monotonic decrease in performance, dropping to 69.2% at $D = 30\%$.

Qualitatively, this performance drop is driven by *generative hallucination*. As the mask expands significantly beyond the object's boundary, the semantic task shifts from "remove this specific object" to "inpaint this large void." For instance, in the top row of the dilation results, as the mask grows to $D = 30\%$, the model ceases to simply remove the people and instead hallucinates a new object (a bench) to fill the empty slab, as its priors dictate that such a space is unlikely to be empty.

**Sensitivity to Erosion (Under-Masking).**    In contrast, the model is highly sensitive to incomplete masking. As shown in Figure 15, a mere $5\%$ erosion causes the Success Rate to fall from 76.4% to 32.5%, eventually reaching $\approx 0\%$ at 20% erosion. This sharp decline is expected in diffusion-based inpainting. when valid object features (e.g., the edge of a dress or a car bumper) remain *outside* the mask, they act as strong conditioning signals. The model interprets these unmasked pixels as ground truth context that must be preserved, forcing it to reconstruct the object rather than remove it.

These findings characterize LRNR as highly forgiving of "lazy" or loose masking but intolerant of partial masking. This suggests a practical guideline for deployment: when using imperfect segmentation tools or manual selection, it is preferable to over-mask rather than under-mask.

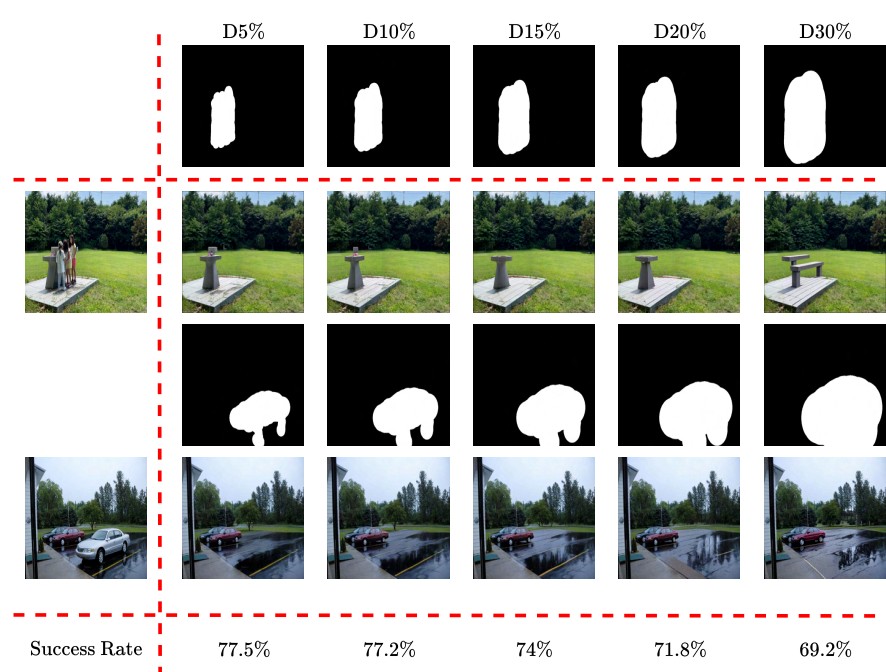

Figure 14: **Ablation study on Mask Dilation.** The model is robust to loose masks, maintaining high success rates ($\sim 77\%$) up to $10\%$ dilation. Extreme dilation ($30\%$) causes the model to hallucinate new objects (e.g., generating a bench) rather than simply removing the target.

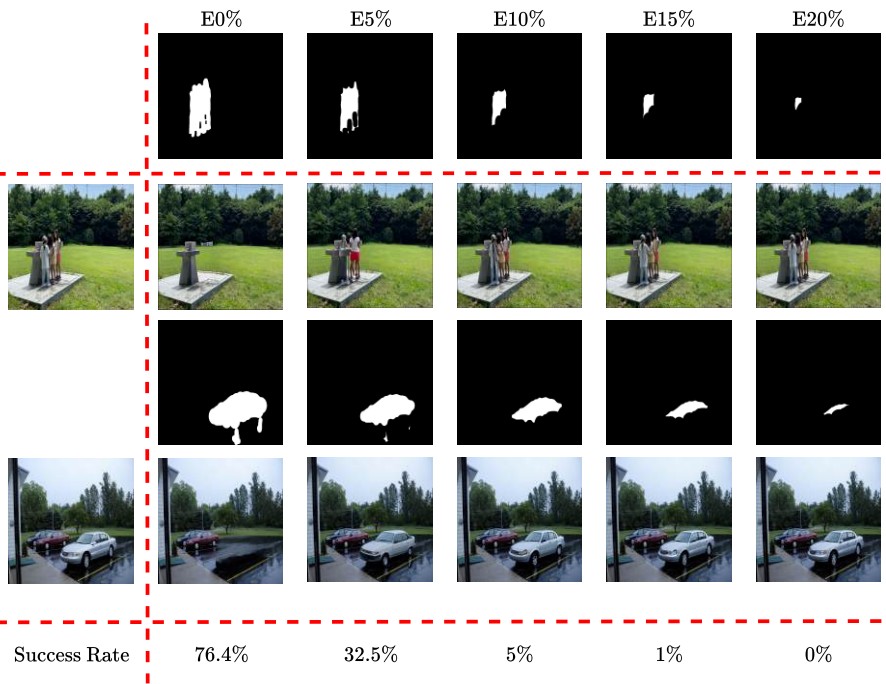

Figure 15: **Ablation study on Mask Erosion.** The model is highly sensitive to incomplete masks. Even $5\%$ erosion results in significant "feature leakage," where unmasked object parts force the model to reconstruct the target, causing success rates to drop sharply to $32.5\%$.

