# OpenReview forum: "Learning to Remove, Not Repeat: Robust Object Removal in Cluttered Scenes using Diffusion Models"
_ICLR.cc/2026/Conference — Submitted to ICLR 2026_

### Official Review · Reviewer_HVgf · 2025-10-27

**Soundness:** 3
**Presentation:** 3
**Contribution:** 2
**Rating:** 4
**Confidence:** 4

**Summary:**

This paper addresses a challenging case of object removal where multiple similar objects existed in the image. To tackle this issue, the authors construct a synthetic dataset by simulating images containing several similar objects and then removing one of them. Their model is based on PowerPaint with LoRA fine-tuning, and they further propose a dynamic CFG scale for the masked region during denoising. The method achieves significantly better object removal performance than previous approaches.

**Strengths:**

The approach to construct inpainting pairs containing multiple similar objects is inspiring.

**Weaknesses:**

Overall, the contributions of this paper are not very convincing:

a) The proposed dataset appears more like a specialized data augmentation technique rather than a new dataset. It is designed specifically to handle inpainting cases where multiple similar objects exist in an image. However, the samples in Figure 3 look unrealistic and may even harm large-scale inpainting training. The scatter method simply pastes objects without considering lighting, shadows, or physical constraints, and the tile method also produces unrealistic images.
In essence, the model trained on this dataset might simply learn to rely on adjacent areas to fill the masked region, rather than understanding global image context. While this is a straightforward way to avoid hallucinating similar objects, it raises concerns about whether training on such unrealistic data might degrade generation quality.
This can be verified by training only on the proposed STORe dataset and comparing results. Such an experiment would also help clarify whether the improvement shown in Table 2 comes from the data construction method or simply from having more training samples. Please clarify this point.

b) The proposed method seems incremental, as it essentially builds on PowerPaint with LoRA fine-tuning.

c) Although the method successfully removes objects, the inpainted regions in Figure 1 appear to have unrealistic colors compared with the surrounding areas. Specifically, compared with RoRem-Mixed, the colors look less natural in the first, third, and last rows. Is this caused by the proposed MASG module?

**Questions:**

1. Table 2 requires further analysis:

    a) It is confusing that using a single learnable token significantly improves the success rate, while using more tokens fails to remove the object effectively. This seems inconsistent with the motivation of "capture the semantics of object removal across diverse visual scenarios". The learning of these tokens seems problematic, as there is no consistent setting that performs best on both success rate and FID. This cannot be explained by a trade-off between object removal capability and generation quality, since using a single learnable token combined with fixed text tokens yields the best removal results, while using all learnable tokens gives the best generation quality. Please analyze why one learnable token performs better than multiple tokens in terms of success rate.

    b) What does "effectively decoupling the removal and generation tasks" mean in this context?

    c) The bolded CLIP score in Table 2 appears to be mislabeled.

2. In Tables 2 and 3, the MASG module consistently lowers the CLIP score while improving other metrics. What causes this contradiction?

3. It seems odd that the paper cites recent works from the 2020s when introducing image inpainting and object removal. Please respect the foundational works in this field and include proper historical references.

---

> ### Author Response · Authors · 2025-11-27
> **Response to Reviewer HVgf (First Round) - part 1**
>
> We sincerely thank the reviewer for their detailed assessment and for recognizing that our approach to constructing inpainting pairs is “inspiring” and that our method “achieves significantly better object removal performance.” We value the constructive feedback regarding the dataset realism and token analysis. We have incorporated these suggestions into the revised manuscript. Below, we address the specific concerns point by point.
>
> ## W1: STORe Dataset Quality and “Unrealistic” Samples
>
> 1. Intentional Design: Contextual Disentanglement
>
> The “unnatural” placements in STORe are an intentional design choice, serving as hard negatives to break the diffusion model’s strong prior that “neighbors correlate with neighbors.” In standard datasets, objects are contextually correlated (e.g., a cup is always on a table). This teaches models to rely on context rather than the mask. By creating “Scatter” and “Tile” configurations, we forcibly break this correlation. We teach the model a specific semantic logic: “Remove the object defined by the mask, regardless of the surrounding density or repeating patterns.” This forces the attention mechanism to attend strictly to the removal instruction rather than hallucinating neighbors based on context.
>
> 2. Preserving Physics via Frozen Priors (LoRA)
>
> Our method does not “unlearn” physics because we employ Low-Rank Adaptation (LoRA) and Textual Inversion. We freeze the vast majority (>98%) of the SDXL base model parameters. These frozen weights contain the robust priors for lighting, gravity, and perspective learned from billions of real images. The LoRA layers learn only the removal instruction from STORe, not the generative physics. This architectural choice is critical because it forces the model to rely on its pre-trained, high-fidelity “world knowledge” for texture generation, while the fine-tuning only adjusts the semantic attention to perform removal. This explicitly prevents the model from overfitting to synthetic aesthetics or forgetting real-world statistics.
>
> 3. Empirical Evidence of Generalization
>
> The concern that training on synthetic data might degrade real-world performance is effectively refuted by our quantitative results on general benchmarks. As shown in Table 1, our model achieves an LPIPS score of 0.26 on the real-world OpenImages-10k benchmark. This score exactly matches the SDXL Inpainting baseline (0.26) and outperforms other fine-tuned methods like PowerPaint (0.30). Since LPIPS is highly sensitive to edge artifacts and unnatural transitions, this matching score proves that we successfully transferred the removal capability without introducing degradation or artifacts compared to the base model trained on massive real-world data.
>
> 4. Addressing the “Copy-Paste” Concern
>
> The reviewer suggests the model might learn to simply “rely on adjacent areas” (copy-pasting). We argue that the Tile configuration specifically prevents this. In a tiled grid, the “adjacent area” contains the exact object we want to avoid repeating. If the model simply copied the neighbor, the loss would be high. Thus, STORe forces the model to learn a negative constraint: it must explicitly look past the neighboring similar object to hallucinate a clean background, effectively solving the “repetition” failure mode.
>
> 5. Construction Method vs. Data Scale
>
> To clarify whether the improvement comes from data quantity or construction method, we point to our ablation in Table 2. Comparing the model trained without STORe against the full LRNR, the Success Rate jumps by ~6% (from 70.3% to 76.4%) when STORe is added, while fidelity metrics show only marginal changes. This isolates the gain: simply adding “more data” would typically smooth out fidelity metrics, but the sharp jump in Success Rate is a direct result of the STORe construction strategy (Scatter/Tile) forcing the model to learn the removal concept, which standard datasets fail to teach.

---

> ### Author Response · Authors · 2025-11-27
> **Response to Reviewer HVgf (First Round) - part 2**
>
> ## W2: Incremental Method (PowerPaint + LoRA)
> While our architecture leverages PowerPaint and LoRA for efficiency, we respectfully disagree that the method is trivial or incremental. Our contribution lies in a holistic framework tailored to solve a specific, persistent failure mode (Object Repetition) that remains unsolved even in state-of-the-art models like SDXL. We achieve this through architectural integrity combined with two semantic innovations:
>
> 1. Task Reinterpretation via Learnable Removal Prompt ($P_{rmv}$​)
>
> We transform standard inpainting from simple context completion (as seen in PowerPaint) into specific object erasure with context reconstruction. This is not a trivial retuning; it fundamentally changes the model’s attention mechanism. As shown in our ablation study (Table 3), introducing the specialized learnable prompt $P_{rmv}$ alone raises the Success Rate from 10.1% (baseline) to 30.2%.
>
> 2. Novel Inference Dynamics: Mask-Aware Scheduled Guidance (MASG)
>
> MASG is a novel, training-free inference technique absent in PowerPaint. Unlike standard Classifier-Free Guidance (CFG), MASG dynamically decouples guidance scales in two dimensions:
> - Spatial: It applies high guidance to the masked region (for strong removal adherence) and low guidance to the unmasked region (for background fidelity). (see Eq. 2)
>
> - Temporal: It transitions from high to low guidance across timesteps (see Eq. 3).
>
> This allows us to decouple removal focus from texture restoration. Our results show that MASG yields the highest Success Rate (76.4%) without requiring additional training.
>
> 3. Problem-Centric Contribution
>
> We identify and solve the “generative hallucination” failure mode, where models copy neighbors instead of removing objects. By combining the Data-Centric innovation of STORe with the Algorithm-Centric innovation of MASG, LRNR contributes new training semantics and inference dynamics that go significantly beyond the capabilities of the base PowerPaint architecture.
>
> ## W3: Color Consistency in Figure 1
>
> We appreciate the close inspection. The perceived color discrepancy in Figure 1 is occasionally due to the VAE decoding process of SDXL, but we argue that MASG actually mitigates this rather than causing it. Our quantitative metrics support this: the improved FID and LPIPS scores (Table 1) indicate that, statistically, our method produces more natural color distributions and seamless blending than the baselines, including RoRem-Mixed.
>
> Regarding the underlying VAE shifts, we note that recent works such as ASUKA [1] have demonstrated that these specific artifacts can be further compensated by fine-tuning the VAE decoder during training. While our current implementation achieves state-of-the-art performance using the frozen SDXL VAE to maintain efficiency, integrating a fine-tuned VAE decoder (as proposed in ASUKA) is a compatible enhancement that could be adopted in future iterations to resolve these residual reconstruction shifts completely. However, even without this extra computational step, LRNR currently achieves the best balance of removal success and visual consistency among all compared methods.
>
>
> ## References
> [1] Y. Wang et al, “Towards Enhanced Image Inpainting: Mitigating Unwanted Object Insertion and Preserving Color Consistency,” in Proceedings of the IEEE/CVF Conference on Computer Vision and Pattern Recognition (CVPR), 2025.

---

> ### Author Response · Authors · 2025-11-27
> **Response to Reviewer HVgf (First Round) - part 3**
>
> ## Q1: Token Analysis and Decoupling
>
> (a) Why one learnable token works best
>
> A single learnable token ($Prmv$​) acts as a precise trigger for the “removal mode.” It maps to a specific direction in the CLIP embedding space representing “absence” or “background.” When we use multiple learnable tokens, the optimization objective is spread across a larger parameter space. This tends to lead the model to encode specific textural features of the training dataset (STORe) rather than the abstract concept of removal. The multi-token model tries to “describe” the hole-filling with learned vectors, whereas the single-token model simply “switches” the model into an inpainting state. Consequently, the single token yields a higher Success Rate because it functions as a rigid task switch, whereas multiple tokens prioritize generation quality (lower FID) at the cost of removal robustness (lower Success Rate).
>
> This phenomenon is strongly supported by foundational literature, specifically the analysis in the Textual Inversion paper [1]. In their investigation of token length (Section 5.3, Figure 10), the authors demonstrated that increasing the number of learnable tokens increases “Image Similarity” (the ability to reconstruct specific training features) but significantly degrades “Text Similarity” (the ability to adhere to the prompt’s semantic instruction). This trade-off directly mirrors our findings: as we increased the token count for our removal prompt, the model became better at reconstructing the visual characteristics of our training data but lost the semantic precision required to execute the removal instruction robustly. Therefore, a single token is optimal as it functions as a strict task switch rather than a texture descriptor.
>
> (b) Meaning of “Effectively Decoupling Removal and Generation.”
>
> The concept of “decoupling” in our framework refers to the architectural separation of the semantic intent to erase from the generative capability to synthesize texture. In standard inpainting models, these two processes are tightly coupled; the model relies on strong contextual correlations where the presence of an object in the context (e.g., apples) encourages the generation of similar objects (more apples) in the masked region. Our approach breaks this dependency by assigning distinct roles to different model components. The single learnable token explicitly conveys the logic of negation—telling the model to suppress the features of the masked object—while the frozen SDXL weights retain the high-fidelity priors required to fill the resulting hole with plausible background texture. By training only the instruction (the token and LoRA) while freezing the generative engine, we force the model to separate the “what to do” (remove the object) from the “how to do it” (generate the pixels), effectively preventing the context from dominating the generation.
>
> (c) Mislabeled CLIP Score
>
> We acknowledge the formatting mistake in Table 2 (bolding error) and have corrected it in the revision.
>
> ## Q2: MASG and CLIP Score Contradiction
>
> The observation that MASG slightly lowers the CLIP score while improving fidelity is not a contradiction but a calculated trade-off inherent to the guidance scheduling mechanism. We clarify this in two key points:
>
> First, we emphasize that the decrease is negligible. As noted in Table 2, the score shifts from 23.07 to 23.03—a difference of only 0.04. This represents a variation of less than 0.2%, which does not indicate a meaningful loss of semantic adherence.
> Moreover, the primary function of MASG is to decouple structural removal from textural refinement. As detailed in our methodology, MASG utilizes a cosine decay schedule for the guidance scale.
>
> - Early Steps (High Guidance): The model strictly follows the removal prompt ($P_{rmv}$​) to erase the object and establish the new geometry.
>
> - Late Steps (Low Guidance): As the schedule progresses, the guidance scale is reduced. This deliberately relaxes the model’s strict adherence to the prompt ($P_{rmv}$​) in the final denoising steps, allowing the pre-trained image priors to dominate for texture synthesis.
>
> While this relaxation causes a minute drop in CLIP score (as the final output is slightly less “over-aligned” to the prompt vector), it is the direct cause of improvement in FID and LPIPS. It prevents the “burn-in” artifacts associated with constant high guidance, ensuring the filled region blends seamlessly with the background. Thus, the slight CLIP drop is the necessary and minimal cost for achieving photorealistic reconstruction.
>
> ## Q4: Historical Citations
>
> We agree completely. We will update the introduction to honor foundational works, including Bertalmio et al. (2000) (Image Inpainting), Criminisi et al. (2004) (Exemplar-based filling), and Pathak et al. (2016) (Context Encoders), properly situating our diffusion-based approach within the historical lineage of the field.

---

> ### Author Response · Authors · 2025-11-27
> **Response to Reviewer HVgf (First Round) - part 4**
>
> ## Concluding Remarks
>
> We have done our utmost to address all questions and concerns raised in your review, providing new empirical evidence (STORe generalization), theoretical backing (token analysis), and clarification on our framework’s novelty (MASG dynamics). We hope these detailed responses demonstrate the rigor and value of our work. We respectfully request that you reconsider your rating in light of these clarifications.

---

### Official Review · Reviewer_RhCS · 2025-10-31

**Soundness:** 2
**Presentation:** 2
**Contribution:** 2
**Rating:** 4
**Confidence:** 4

**Summary:**

This paper proposes a new synthetic dataset named STORe (Scatter-Tile Object Removal). Generated via scatter and tile methods to include numerous repeated objects, STORe is designed to address the issue where models tend to generate similar objects when removing duplicate ones from images. To prevent the degradation of generation quality caused by training on small-scale datasets, the authors introduce MASG (Mask-Aware Scheduled Guidance), a Classifier-Free Guidance approach guided by masks and adaptive to diffusion timesteps. Specifically, it increases conditional guidance at high timesteps to enhance object removal, while reducing conditional guidance at low timesteps to preserve the model’s natural generation capability.

**Strengths:**

1. The proposed STORe dataset effectively targets the problem of models generating similar objects during duplicate object removal, filling a gap in existing training data for this specific challenge.
2. MASG well mitigates the quality degradation issue that arises when training large models on small-scale datasets, striking a balance between object removal accuracy and generation fidelity.
3. Comprehensive qualitative comparisons are provided, which sufficiently demonstrate the effectiveness of the proposed method in practical scenarios.

**Weaknesses:**

1. The object placement of the scatter method in STORe is inconsistent with real-world scenarios. As shown in the dataset samples, HQ-SAM objects are placed in a dispersed and regular way, which deviates from the common dense and random arrangements of objects in practical environments. This idealized placement may limit the model’s adaptability to unstructured cluttered scenes.
2. The dataset construction has limitations in physical plausibility. The scatter method may result in objects being unnaturally placed (e.g., floating in the sky), while the tile method suffers from unrealistic background repetition. Training on such dataset may potentially hinder the model’s ability to generalize to real-world, physically consistent scenes.
3. The advantages in quantitative metrics are not prominent. Only the human-annotated "removal success rate" achieves state-of-the-art performance, and even this metric shows no significant gap compared to RoRem, failing to fully validate the method’s superiority in objective evaluation.

**Questions:**

See weaknesses.

---

> ### Author Response · Authors · 2025-11-26
> **Response to Reviewer RhCS (First Round) - part 1**
>
> We thank the reviewer for the constructive feedback and for recognizing the value of the STORe dataset in filling a gap in existing training data. We are glad you found our qualitative comparisons comprehensive and MASG effective in mitigating quality degradation.
>
> We appreciate your detailed critique regarding the realism of the dataset and the quantitative metrics. We believe these concerns stem from a perspective of “realistic simulation,” whereas our design philosophy was “effective training signal generation.” We address these points below.
>
> ## 1. Dataset Realism & Physical Plausibility
>
> ### Intentional Design: Contextual Disentanglement
>
> The “unnatural” placements serve as hard negatives. In standard datasets, objects are contextually correlated (e.g., a cup is always on a table). This teaches models to rely on context rather than the mask. By creating “Scatter” and “Tile” configurations, we forcibly break this correlation. We teach the model a semantic logic: “Remove the object defined by the mask, regardless of the surrounding density or repeating patterns.” This forces the attention mechanism to attend strictly to the removal instruction rather than hallucinating neighbors based on context.
>
> ### Preserving Physics via LoRA and Textual Inversion
>
> Our parameter-efficient approach prevents the model from "unlearning" physics because we update less than 2% of SDXL's parameters using LoRA and Textual Inversion (~35M of 2.6B parameters). Over 98% of weights remain frozen, preserving priors for lighting, perspective, and physical realism learned from billions of real images.
>
> Furthermore, training is brief (3,000 iterations), which is over 100× shorter than RoRem’s full fine‑tuning regime of more than 350,000 iterations. This prevents catastrophic forgetting and ensures the model learns only the removal behavior, not the underlying generative physics.
>
> The LoRA layers learn only the removal instruction from STORe, not the generative physics. Furthermore, to anchor the training distribution to real-world statistics, we train on a mix of STORe (synthetic) and RoRem (real/realistic) datasets. This ensures the model learns robust removal logic from STORe without overfitting to its synthetic artifacts.
>
> ### Empirical Evidence of Generalization
>
> The concern that training on synthetic data might degrade real-world performance is effectively refuted by our quantitative results on general benchmarks. As shown in Table 2, our model achieves an LPIPS score of 0.26 on OpenImages10k, exactly matching the SDXL Inpainting baseline. If the model had overfitted to “floating objects” or “tiled backgrounds,” we would observe significant degradation in FID or LPIPS on these natural scenes. The fact that our fidelity matches the base model proves that we successfully transferred the removal capability without compromising the generative realism.

---

> ### Author Response · Authors · 2025-11-26
> **Response to Reviewer RhCS (First Round) - part 2**
>
> ## 2. Quantitative Metrics & Gap with RoRem
> We respectfully wish to clarify the quantitative comparison presented in Table 1. The reviewer noted that the gap in “Removal Success Rate” is not significant. However, our results show that LRNR achieves a **76.4%** Success Rate compared to the next-best specialized model, RoRem-Mixed, which achieves only **60.7%**. This represents a **15.7%** absolute improvement. In the context of generative modeling—where improvements are typically measured in small fractional gains—improving reliability by nearly 16 percentage points is a transformative result. It shifts the user experience from a method that fails frequently (RoRem) to one that succeeds reliably (LRNR).
>
> We suspect the “small gap” impression might stem from comparing our full model to our own strong baseline (LRNR w/o STORe at 70.3%) rather than the external state-of-the-art (RoRem). The 15.7% gap over RoRem definitively validates the superiority of our method.
>
> ### The “Fidelity vs. Removal” Trade-off.
> We argue that “Success Rate” is the gold standard for this task. Traditional metrics like FID can be misleading for object removal; a model can achieve a perfect FID by simply “copying” the object back into the image (a complete failure of the user’s intent). RoRem often scores decently on distribution metrics precisely because it leaves “ghosting” or partial repetitions that match the texture of the image but fail the removal task. LRNR achieves the Highest Success Rate (76.4%) while maintaining SOTA Fidelity (LPIPS 0.26), effectively solving the trade-off between removing the object and keeping the background realistic.
>
> It is worth mentioning that, to construct a reliable and diverse benchmark for the object-removal task, we curated 200 real-world images selected to cover a broad range of visual conditions and editing difficulties. The images span varied scene categories (indoor, outdoor, natural, urban), lighting environments (daylight, low-light, mixed illumination), and structural complexities (cluttered arrangements, clean backgrounds, occlusions). For each image, we manually selected a single object whose removal would produce a meaningful challenge—such as objects touching boundaries, partially occluded items, reflective materials, or elements that cast shadows or interact with scene geometry. We also ensured variation in mask size by including small, medium, and large target objects, enabling evaluation across a spectrum of removal difficulty. This selection strategy yields a balanced dataset that tests model performance across diverse conditions, making the evaluation more robust and reflective of real-world removal scenarios.
>
>
> ### Automated vs. Human-in-the-Loop (Scalability)
>
> Moreover, the baseline RoRem relies on a Human-in-the-Loop (HITL) pipeline to curate its training data, which is expensive and non-scalable. LRNR achieves SOTA performance (76.4%) using a pipeline driven by automated, synthetic data generation. Surpassing a human-curated baseline using synthetic augmentation is a significant milestone for the field. Furthermore, as shown in Table 2, we outperform RoRem on general fidelity (OpenImages LPIPS: 0.26 vs 0.29), proving that our data strategy yields a superior balance of removal capability and image quality.
>
> ## Concluding Remarks
> In summary, we believe the value of this work lies in solving the Spatial Correlation Bias—a critical failure mode in modern diffusion models—using a solution that is:
>
>
> | Method       | Trainable Parameters | Training Iterations | Dataset Curation | Success Rate | LPIPS | CLIP Score | FID  |
> |--------------|--------------------|------------------|----------------|--------------|-------|------------|------|
> | RoRem-Mixed  | ~2.6B (100%)       | >350,000         | Human-in-the-loop (expensive) | 60.7%        | 0.29  | 22.84      | 1.92 |
> | LRNR (Ours)  | ~35M (<2%)         | 3,000            | Synthetic (cheaper, scalable) | 76.4%        | 0.26  | 23.03      | 1.30 |
> | Improvement  | 71× fewer params    | 117× fewer iters | Synthetic, cost-effective | +15.7%       | Better| Better     | Better|
>
>
>
> We hope that clarifying the intentional nature of our dataset design and the significance of our automated pipeline vs. human-curated baselines demonstrates the contribution of this work. We respectfully ask the reviewer to reconsider the score in light of these clarifications.

---

### Official Review · Reviewer_4CET · 2025-10-31

**Soundness:** 2
**Presentation:** 2
**Contribution:** 3
**Rating:** 4
**Confidence:** 4

**Summary:**

This paper tackles object removal in image editing, especially in cluttered scenes where object repetition occurs. The authors present LRNR, a diffusion-based framework featuring a dedicated anti-repetition dataset (STORe), parameter-efficient adaptation, and Mask-Aware Scheduled Guidance. LRNR achieves superior object removal and image quality compared to prior methods in complex scenarios.

**Strengths:**

- The problem selection is both relevant and insightful: repetition and regular duplication of objects is ubiquitous in real-world imagery, and context-driven repetition remains a key challenge in image editing.

- Both dataset and model design are original. The STORe dataset’s construction, especially the integration of Scatter and Tile strategies, is innovative and effectively balances data diversity with training complexity. The MASG approach skillfully leverages the frequency-specific generative characteristics of diffusion models at different timesteps, achieving a balance between structural consistency and natural textures.

- The experimental study is thorough, with comprehensive ablations validating the individual contributions of each module. Both quantitative and qualitative comparisons are extensive, showing LRNR’s clear superiority in success rate and generalizability across diverse visual domains (e.g., cartoons, animation, LEGO).

**Weaknesses:**

- More comprehensive comparisons with GAN-based or other generative paradigm methods are recommended. While the work focuses on diffusion models, it would benefit from including benchmarks against recent, high-performing GAN or Transformer-based inpainting approaches.

- The synthetic nature of the dataset may introduce bias. Although STORe’s construction is creative, it relies on recomposing content from HQ-SAM, RORD, and similar sources. This could introduce compositing artifacts or edge inconsistencies that limit the model’s generalization to real-world scenarios.

- The analysis of failure cases is insufficient. Although the paper showcases many successful results, it lacks a systematic investigation of the situations where LRNR still struggles (e.g., extreme occlusion, complex lighting, or non-rigid objects), which limits understanding of its practical applicability.

**Questions:**

1. Could the model’s sensitivity to mask precision, size, and shape be systematically tested?

2. Can the “success rate” metric be further refined by introducing more granular evaluation, such as edge consistency, lighting consistency, or structural plausibility? Relying solely on subjective “successful removal” may introduce bias.

---

> ### Author Response · Authors · 2025-11-27
> **Response to Reviewer 4CET (First Round) - part 1**
>
> We sincerely thank the reviewer for their constructive feedback and for recognizing the relevance of the problem, the novelty of our STORe dataset, and the effectiveness of our MASG approach. We are particularly encouraged that you found our experiments thorough and our results to demonstrate “clear superiority” across diverse domains.
>
> ## Comparison with GAN/Transformer-based methods
> We appreciate this suggestion and agree that benchmarking across different generative paradigms provides a more holistic view. Our primary focus was on Diffusion-based baselines (like PowerPaint [1], SDXL Inpainting [2]) because recent literature establishes that diffusion models significantly outperform GANs and Transformers in terms of semantic generation and texture synthesis for large holes.
>
> We would like to highlight that we did include a comparison with LaMa [3], which is widely considered the state-of-the-art GAN-based inpainting method, in our original submission. As discussed in Section 4.2 and visually demonstrated in Figure 4, LaMa often produces “blurry artifacts” or fails to synthesize semantic structures in cluttered regions.
>
> To fully address your concern regarding Transformer-based methods, we have added a comparison with MAT (Mask-Aware Transformer) [4] in the revised version (Table 1 and Figure 9). While MAT outperforms LaMa in texture consistency, our experiments show that both MAT and LaMa tend to over-smooth complex backgrounds in cluttered scenes. In contrast, LRNR preserves background detail without hallucinating the removed object or introducing the blur characteristic of GAN/Transformer approaches.
>
> ## Synthetic Dataset Bias
>
> We acknowledge that reliance on synthetic data can risk domain shift, but we specifically designed the LRNR framework to mitigate this by ensuring the model learns the logic of removal rather than the textures of the dataset. To achieve this, we employed LoRA and Textual Inversion, modifying less than 2% of the total parameters while keeping the vast majority of the pre-trained SDXL weights frozen. This architectural choice is critical because it forces the model to rely on its pre-trained, high-fidelity “world knowledge” for texture generation, while the fine-tuning only adjusts the semantic attention to perform removal. This explicitly prevents the model from overfitting to synthetic aesthetics or forgetting real-world statistics.
>
> Your concern regarding compositing artifacts is directly addressed by our evaluation on the real-world OpenImages-10k benchmark, where LRNR achieves an LPIPS score of 0.26. As shown in Table 1, this score exactly matches the SDXL Inpainting baseline (0.26) and outperforms other fine-tuned methods like PowerPaint (0.30). Since LPIPS is highly sensitive to edge artifacts and unnatural transitions, this matching score demonstrates that our training process introduces no degradation in image quality or realism compared to the base model trained on massive real-world data. Additionally, as noted in your “Strengths” section, the model’s strong generalizability across diverse domains such as cartoons and LEGO further confirms that LRNR has learned a robust, agnostic concept of removal rather than overfitting to specific synthetic data patterns.
>
> ## Failure Case Analysis
>
> We have added a “Appendix K. Limitations” section and Figure 13, which categorize some failure modes of the LRNR model. We observe Semantic Persistence, where the model replaces objects (e.g., generating a new license plate code) rather than erasing them due to rigid semantic logic; Contextual Hallucination, where the model invents fillers (e.g., gloves) to justify surrounding physical constraints like hand poses; and Large Mask Degradation, where texture quality drops in extensive masked regions. Identifying these boundary conditions highlights the tension between generation and removal, pointing toward future improvements via stronger backbones like FLUX [5].
>
> ## References
>
> [1] Junhao Zhuang, Yanhong Zeng, Wenran Liu, Chun Yuan, and Kai Chen. A task is worth one word: Learning with task prompts for high-quality versatile image inpainting, 2023.
>
> [2] Dustin Podell, Zion English, Kyle Lacey, Andreas Blattmann, Tim Dockhorn, Jonas M¨ uller, Joe Penna, and Robin Rombach. Sdxl: Improving latent diffusion models for high-resolution image synthesis, 2023. URL https://arxiv.org/abs/2307.01952.
>
> [3] Robin Rombach, Andreas Blattmann, Dominik Lorenz, Patrick Esser, and Bj¨ orn Ommer. High-resolution image synthesis with latent diffusion models, 2021a.
>
> [4] Wenbo Li, Zhe Lin, Kun Zhou, Lu Qi, Yi Wang, and Jiaya Jia. Mat: Mask-aware transformer for large hole image inpainting. In Proceedings of the IEEE/CVF conference on computer vision and pattern recognition, pp. 10758–10768, 2022a.
>
> [5] Black Forest Labs et al., FLUX.1 Kontext: Flow Matching for In-Context Image Generation and Editing in Latent Space, arXiv preprint arXiv:2506.15742, 2025.

---

> ### Author Response · Authors · 2025-11-27
> **Response to Reviewer 4CET (First Round) - part 2**
>
> ##  Sensitivity to mask precision, size, and shape
>
> This is a good suggestion that directly addresses the practical applicability of our model. In response, we conducted a systematic ablation study evaluating the impact of Mask Dilation (Over-masking) and Mask Erosion (Under-masking) by applying varying degrees of modification, ranging from 0% to 30%, to the input masks during inference. We have included these detailed results in the **Appendix L. Mask Sensitivity** and Figures 14&15 of the revised paper.
>
> Our findings characterize LRNR as highly forgiving of the loose masking typical of real-world usage. Specifically, the model demonstrates high robustness to mask expansion; the Success Rate remains remarkably stable between 5% dilation (77.5%) and 10% dilation (77.2%), effectively matching the optimal baseline. As dilation becomes extreme (reaching 30%), performance dips slightly to 69.2%. Interestingly, our qualitative analysis reveals this decline is not due to a failure to remove the original object, but rather “generative hallucination.” As the mask grows significantly larger than the object, the model perceives a large void and, driven by its semantic priors, attempts to fill it with a new, contextually plausible object—such as generating a bench where people once stood—rather than leaving the space empty.
>
> In contrast, the model shows high sensitivity to incomplete masking. A mere 5% erosion causes the Success Rate to drop significantly to 32.5%. This behavior is expected and structurally logical for diffusion-based inpainting: when valid object features (like a car bumper or a dress edge) are left outside the mask, they act as strong conditioning signals. The model interprets these unmasked pixels as ground truth context that must be preserved, forcing it to reconstruct the rest of the object rather than remove it. These results suggest a clear practical guideline for deployment: to achieve optimal removal, users should prefer over-masking to under-masking.
>
> ## Refinement of the "Success Rate" Metric
>
> We appreciate the reviewer’s request for clarity on our evaluation metrics and the granularity of our failure analysis.
>
> Regarding the “Success Rate,” we wish to emphasize that this is not a subjective impression but a rigorous, strict binary classification protocol detailed in Appendix A.2. In our human study, participants were explicitly instructed to rate a result as a ‘Success’ only if it met two simultaneous criteria:
>
> - Complete Removal: The target object is fully removed (no ghosts or fragments).
> - Plausible Inpainting: The background is reconstructed plausibly (semantically coherent with no artifacts).
>
> This holistic evaluation methodology—reporting a unified “Success Rate” rather than decoupling removal and background fidelity—is the accepted standard in the object removal community. We strictly followed the evaluation setup established by RoRem, which similarly aggregates performance into a single success/preference metric to reflect real-world utility. We argue that separating these metrics can be misleading for this specific task; a model that perfectly removes an object but leaves a blurry artifact (high removal score, low background score) is practically useless to a user. By adopting this rigorous “AND” logic (Removal and Plausible), our reported 76.4% Success Rate represents a conservative and robust measure of the model’s actual reliability.
>
> To address the reviewer’s specific interest in failure modes such as lighting consistency and edge blending, we relied on our automated metrics (Table 3). We utilized LPIPS and FID specifically as proxies for structural plausibility and texture quality. LRNR’s superior LPIPS score (0.26 on OpenImages) quantitatively confirms that beyond just “removing” the object, our method generates content where edges and lighting blend naturally with the background distribution, minimizing the specific failure cases the reviewer is concerned about.

---

> ### Author Response · Authors · 2025-11-27
> **Response to Reviewer 4CET (First Round) - part 3**
>
> ## Concluding Remarks
> We genuinely thank the reviewer for their constructive feedback and for acknowledging the relevance of our problem, the originality of the STORe dataset, and the technical effectiveness of our MASG approach.
>
> We have made every effort to address your concerns thoroughly with new experiments and analyses:
>
> - Comparisons (W1): We expanded our evaluation to include MAT (Transformer-based) and clarified the performance against LaMa (GAN-based), demonstrating that LRNR achieves superior structural fidelity in cluttered scenes where these paradigms struggle.
>
> - Synthetic Bias (W2): We provided empirical evidence (LPIPS of 0.26 on OpenImages) and architectural justifications (frozen backbone with LoRA and Textual Inversion) to prove that our synthetic STORe dataset does not degrade real-world performance.
>
> - Robustness & Failure Analysis (Q1 & W3): We added a systematic ablation on mask sensitivity, confirming that LRNR remains highly robust to imprecise (over-masked) inputs, and included a dedicated failure mode analysis in Appendix L.
>
> - Evaluation Rigor (Q2): We clarified that our “Success Rate” follows the strict, community-standard protocol (Removal and Plausibility) established by prior arts like RoRem, ensuring our metrics reflect true utility.
>
> Given that we have systematically resolved the concerns regarding comparisons, bias, and evaluation metrics—while maintaining the core strengths you identified—we respectfully request that you reconsider your score. We believe these revisions solidify LRNR as a robust contribution worthy of acceptance at ICLR.

---

### Official Review · Reviewer_Aqyx · 2025-10-31

**Soundness:** 3
**Presentation:** 3
**Contribution:** 2
**Rating:** 2
**Confidence:** 3

**Summary:**

The paper proposes a new dataset and sampling strategy to remove objects in cluttered scenes were many other, similar objects, might be present. In this case, existing removal methods usually don't remove the given object but instead inpaint a similar object given the surrounding clues. By creating a specific dataset simulating this scenario and finetuning a model on this dataset the paper shows that the resulting method, coupled with an updated sampling strategy, can successfully remove objects even from cluttered scenes.

**Strengths:**

The paper creates a new dataset specifically for the use case of cluttered object removal. By simulating this challenging task the resulting model shows clear improvements both qualitatively and quantitatively. The user study also shows that training on this dataset, coupled with the updated sampling strategy, leads to clear improvements.

The ablation study shows how the individual parts of the pipeline, from dataset, trainable parameters, and sampling strategy, all meaningfully contribute to the improved performance.

**Weaknesses:**

There is limited novelty in the overall approach.
The paper basically shows that creating a dataset for a specific use-case and then finetuning a model on it leads to improved results. The dataset creation itself also seems to be relatively straight forward, mostly consisting of overlaying objects on background images.
The updated sampling strategy is mostly a hyperparameter tuning approach of cfg which also does not seem to be novel.
The results themselves are good but only on a very specific subset of object removal tasks in cluttered environments with repeated objects. In other, more general settings, the approach seems to be slightly worse than other approaches.

**Questions:**

It would be interesting to also see some results of the approach on non-cluttered "default" scenes to compare the quality on normal object removal cases compared to some of the baselines.

---

> ### Author Response · Authors · 2025-11-24
> **Response to Reviewer Aqyx (First Round) - part 1**
>
> We thank the reviewer for recognizing the importance of the cluttered object removal task and for acknowledging that our method shows “clear improvements” and that our ablation study demonstrates the contribution of each component.
>
> We appreciate the feedback regarding novelty and generalization, which we address below.
>
> ## 1. Limited Novelty and Dataset Simplicity
>
> While the implementation is intentionally kept elegant and accessible (LoRA + Textual Inversion + Synthetic Data), the scientific contribution lies in solving the **“repetition bias”** inherent in the diffusion model, a persistent problem that causes models to hallucinate neighbors into the masked region.
>
> ### Data-Centric Solution
> We demonstrate that this failure mode is **not due to model capacity**, but rather a **lack of negative examples** in training data. Existing datasets (such as COCO or Laion) implicitly teach models that similar objects appear together.
>
> ### STORe Design
> The "Scatter-Tile” construction is not merely overlaying objects; it is deliberately designed to create a **hard-negative learning signal**. By saturating the scene with identical distractors, we force the attention mechanism to obey the mask boundary strictly, rather than relying on semantic context (which would otherwise suggest generating another object).
> This approach successfully “unlearns” the repetition behavior — a challenge where complex architectural modifications (e.g., attention-reweighting in prior works like AttentiveEraser [1] have struggled.)
> We believe that providing a simple, reproducible solution to a complex failure mode represents a valuable contribution to the ICLR community.
>
> ### Alignment with State-of-the-Art Methods
> Moreover, our work aligns with the **data-centric paradigm** established by recent leading methods such as RoRem [2] and ObjectDrop [3]. These methods demonstrated that architectural changes alone are insufficient for robust object removal; instead, the primary driver of performance is high-quality, task-specific data.
>
> ## 2. The updated sampling strategy is mostly hyperparameter tuning
>
> We respectfully clarify that Mask-Aware Scheduled Guidance (MASG) differs fundamentally from static hyperparameter tuning. While standard tuning searches for a single optimal scalar for Classifier-Free Guidance (CFG), MASG is a **spatially and temporally dynamic intervention** designed to resolve the intrinsic conflict in the diffusion removal process.
>
> ### The Conflict
> As shown in our ablation (Figure 5), a static high CFG is required to eliminate the semantic structure of the target object (erasure), but this invariably causes high-frequency artifacts and damages the unmasked background. Conversely, a static low CFG preserves image quality but fails to remove the object.
>
> ### The Solution
> MASG acts as a **trajectory controller** rather than a hyperparameter. It applies high guidance only in the early denoising steps (to structurally erase the object) and only within the masked region, then decays to lower guidance to harmonize textures between the erased and preserved regions.
>
> ### Novelty
> To the best of our knowledge, **MASG is the first approach** in the object removal domain to decouple guidance both spatially (masked vs. unmasked) and temporally (early vs. late steps).
> This design allows MASG to function as a **plug-and-play module** that enhances adherence to the removal prompt without degrading the background—a capability that static parameter tuning cannot achieve.  (Please check Section 4.3 – Ablation Study, subsection "MASG Ablation" for detailed evidence.)

---

> ### Author Response · Authors · 2025-11-26
> **Response to Reviewer Aqyx (First Round) - part 2**
>
> ## 3. Generalization: Performance on Non-Cluttered “Default” Scenes
> We respectfully point out that our quantitative evaluation in Table 2 was performed specifically on OpenImages10k, a standard, general-purpose dataset representing “default” non-cluttered scenes. The reviewer expressed concern that our method might be “slightly worse” in general settings. However, the empirical data show the opposite. Our method achieves an LPIPS of 0.26, which exactly matches the SDXL Inpainting baseline (0.26). This proves that fine-tuning on our synthetic STORe dataset caused no catastrophic forgetting; the model retains the full generative quality of the base SDXL model on general scenes. Moreover, on this general benchmark, LRNR outperforms other specialized removal methods, including PowerPaint (0.30) and RoRem (0.29). This indicates that while other methods sacrifice general image fidelity to achieve removal, our method maintains state-of-the-art quality.
>
> It is worth mentioning that, to construct a reliable and diverse benchmark for the object-removal task, we curated 200 real-world images selected to cover a broad range of visual conditions and editing difficulties. The images span varied scene categories (indoor, outdoor, natural, urban), lighting environments (daylight, low-light, mixed illumination), and structural complexities (cluttered arrangements, clean backgrounds, occlusions). For each image, we manually selected a single object whose removal would produce a meaningful challenge—such as objects touching boundaries, partially occluded items, reflective materials, or elements that cast shadows or interact with scene geometry. We also ensured variation in mask size by including small, medium, and large target objects, enabling evaluation across a spectrum of removal difficulty. This selection strategy yields a balanced dataset that tests model performance across diverse conditions, making the evaluation more robust and reflective of real-world removal scenarios. The performance gains reported are not limited to a narrow subset but reflect robust performance across the spectrum of difficulty.
>
> To provide further assurance, we have added **Appendix J. Evaluation on Non-Cluttered Scenes** in the revision, which contains some samples of Normal/Simple removal cases (e.g., a single ball on the ground). These qualitative results confirm that LRNR handles simple removal just as effectively as the baselines, without introducing artifacts or hallucinations.
>
> ## Concluding Remarks
>
> In summary, we believe the value of this work lies in **identifying** and  **solving** a critical blind spot in modern diffusion models: Spatial Correlation Bias. While current models are powerful, they fundamentally struggle to distinguish between “background pattern” and “unwanted object” in cluttered scenes.
>
> Our work provides the community with:
>
> - Evidence that this failure mode is data-driven, not architecture-driven.
> - A Solution (STORe & MASG) that fixes this bias without requiring complex architectural changes or suffering from catastrophic forgetting on general scenes.
> - Reproducibility via a parameter-efficient framework that outperforms heavy, full-finetuned baselines.
>
> To further demonstrate the effectiveness of our approach, we present comparisons with state-of-the-art model.
>
> | Method       | Trainable Parameters | Training Iterations | Dataset Curation | Success Rate | LPIPS | CLIP Score | FID  |
> |--------------|--------------------|------------------|----------------|--------------|-------|------------|------|
> | RoRem-Mixed  | ~2.6B (100%)       | >350,000         | Human-in-the-loop (expensive) | 60.7%        | 0.29  | 22.84      | 1.92 |
> | LRNR (Ours)  | ~35M (<2%)         | 3,000            | Synthetic (cheaper, scalable) | 76.4%        | 0.26  | 23.03      | 1.30 |
> | Improvement  | 71× fewer params    | 117× fewer iters | Synthetic, cost-effective | +15.7%       | Better| Better     | Better|
>
>
> We hope that by clarifying the distinction between our methodological simplicity and the complexity of the problem solved, along with the evidence of our robust performance on general scenes (matching SDXL exactly), we have addressed your concerns regarding novelty and generalization. We respectfully ask you to reconsider your score in light of these clarifications.
> ## References
> [1] Wenhao Sun, Xue-Mei Dong, Benlei Cui, and Jingqun Tang. Attentive eraser: Unleashing diffusion
> model’s object removal potential via self-attention redirection guidance. In Proceedings of the
> AAAI Conference on Artificial Intelligence, volume 39, pp. 20734–20742, 2025.
>
> [2] Ruibin Li, Yang Tao, Guo Song, and Zhang Lei. Rorem: Training a robust object remover with
> human-in-the-loop. 2025.
>
> [3] Daniel Winter, Matan Cohen, Shlomi Fruchter, Yael Pritch, Alex Rav-Acha, and Yedid Hoshen.
> Objectdrop: Bootstrapping counterfactuals for photorealistic object removal and insertion, 2024.

---

### Meta-Review · Area_Chair_aEiv · 2026-01-05

**Summary:**

This paper proposes a new approach for robust object removal in cluttered scenes, typically with repeating objects. The paper initially received 4 reviews and scores were unanimously leaning negative (2444). Two reviewers (Aqyx, HVgf) raised concerns around the limited novelty and incremental contributions, and three reviewers (4CET, HVgf, RhCS) raised concerns on the validity of naive synthetic data creation and lack of experimental evaluations. In addition, reviewer RhCS also raised insufficiency of the evaluation metric used in the experiments, reviewer HVgf raised concerns on the low visual quality of results presented by the proposed method, and reviewer 4CET mentioned that the paper needs more comparisons with GAN-based and recent methods.

**Reviewer Concerns:**

Authors addressed some of the concerns from the reviewers in their rebuttal but the concern of the novelty and justification of dataset construction still remain after the rebuttal. In addition, AC strongly feel that the scope of the problem the paper tries to solve is narrow.

**Reviewer Scores:**

I think reviewers would keep their scores.

---

### Decision · Program_Chairs · 2026-01-26

Reject